# LLMs as In-Context Meta-Learners for Model and Hyperparameter Selection

## Abstract

Model and hyperparameter selection are critical but challenging in machine learning, typically requiring expert intuition or expensive automated search. We investigate whether large language models (LLMs) can act as in-context meta-learners for this task. By converting each dataset into interpretable metadata, we prompt an LLM to recommend both model families and hyperparameters. We study two prompting strategies: (1) a zero-shot mode relying solely on pretrained knowledge, and (2) a meta-informed mode augmented with examples of models and their performance on past tasks. Across synthetic and real-world benchmarks, we show that LLMs can exploit dataset metadata to recommend competitive models and hyperparameters without search, and that improvements from meta-informed prompting demonstrate their capacity for in-context meta-learning. These results highlight a promising new role for LLMs as lightweight, general-purpose assistants for model selection and hyperparameter optimization.

## 1 Introduction

The performance of machine learning (ML) models hinges on the selection of appropriate algorithms and their hyperparameters. This joint optimization task is commonly referred to as the Combined Algorithm Selection and Hyperparameter optimization (CASH) problem (Thornton et al., 2013; Bergstra & Bengio, 2012; Snoek et al., 2012). Traditionally, practitioners have relied on manual tuning, grid search, or Bayesian optimization techniques (Mockus et al., 1978; Shahriari et al., 2016) to navigate this complex search space. However, these approaches are computationally expensive and demand substantial domain expertise. This creates barriers to entry and limits the scalability of ML applications across diverse domains.

Large language models (LLMs) have recently shown strong capabilities in reasoning, knowledge synthesis, and problem-solving across domains (Wei et al., 2022). As they scale, they exhibit emergent behaviors that enable adaptation to new tasks by reusing prior experience in context (Brown et al., 2020; Dong et al., 2024). These behaviors have been interpreted as a form of in-context meta-learning, with transformers proposed as general-purpose meta-learners (Kirsch et al., 2024) and LLMs studied explicitly in this role (Coda-Forno et al., 2023). Much of this prior work has focused on demonstrating the phenomenon itself, often in synthetic or language-oriented tasks. By contrast, model and hyperparameter selection provides a practical and consequential setting in machine learning where generalization across tasks directly impacts performance and efficiency. If LLMs can transfer knowledge in this context, they may offer a new paradigm for addressing the CASH problem and extend our understanding of their adaptability beyond controlled demonstrations. This research introduces two prompting strategies for leveraging LLMs in model and hyperparameter selection. The *Zero-Shot* strategy relies solely on high-level task metadata, requiring no prior examples. The *Meta-Informed* strategy augments this by incorporating pairs of task metadata and well-performing model configurations from previous tasks, enabling more informed recommendations (Figure 1). Unlike prior work (Zheng et al., 2023; Zhang et al., 2024), our approach operates without iterative validation feedback. It also enables cross-task generalization in the meta-informed case. Importantly, we prompt the LLM to propose complete configurations consisting of both model families and associated hyperparameters, which can then be directly evaluated or integrated into downstream pipelines.

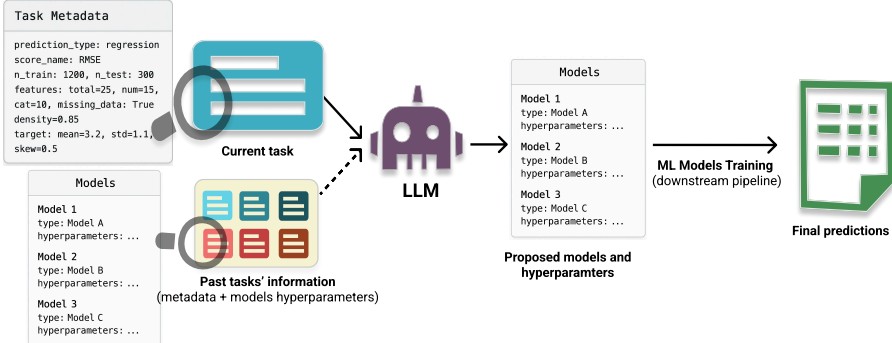

Figure 1: Overview of the method. Each task is represented by metadata, and the LLM outputs model and hyperparameter configurations. The dotted arrow indicates the inclusion of prior-task metadata-configuration pairs in the *meta-informed* setting.

We evaluate both prompting strategies on tabular regression and classification tasks. Results show that LLMs, when properly prompted, can make surprisingly effective recommendations even in zero-shot settings where conventional methods often require extensive experimentation. The meta-informed strategy further improves performance by leveraging prior knowledge, often approaching or matching the quality of expert-guided selections. Taken together, these findings highlight the potential of LLMs as meta-learners in automated machine learning: they can reason about datasets, models, and hyperparameters with minimal tuning, offering a scalable and accessible alternative to traditional search-based or expert-driven workflows. This also complements concurrent applications of LLMs to other stages of the AutoML pipeline such as feature engineering with CAAFE (Hollmann et al., 2023).

The remainder of this paper is structured as follows. Section 2 reviews related work in hyperparameter optimization, meta-learning, and LLM-based methods. Section 3 introduces our formal problem setup and frames CASH as a meta-learning task. Section 4 presents a controlled synthetic experiment that motivates our approach by showing how LLM prompting can capture useful hyperparameter patterns in a simple setting. Section 5 then describes our methodology and evaluates LLM-based prompting strategies on a diverse suite of benchmark datasets. Section 6 discusses broader implications, limitations, and future directions. Finally, Section 7 summarizes our contributions.

## 2 RELATED WORK

**Hyperparameter Optimization.** Early work on hyperparameter optimization (HPO) relied on simple search strategies such as grid search and random search (Bergstra & Bengio, 2012). More sophisticated model-based methods, such as Bayesian optimization (BO), iteratively fit surrogate models to past evaluations and propose promising configurations (Bergstra et al., 2011; Snoek et al., 2012). Subsequent advances introduced multi-fidelity and bandit-based approaches, including Successive Halving (Jamieson & Talwalkar, 2016) and Hyperband (Li et al., 2017), which exploit early stopping to allocate resources efficiently. Later extensions sought to transfer knowledge across related tasks or account for computational budgets, for example through multi-task Bayesian optimization and compute-aware methods (Swersky et al., 2013; Golovin et al., 2017). These methods significantly improved efficiency but still treat each optimization task largely in isolation.

**Meta-Learning for HPO.** To overcome this limitation, meta-learning approaches aim to accelerate HPO by leveraging prior experience across tasks. Transfer Neural Processes (TNP) (Wei et al., 2021), for example, incorporate meta-knowledge such as surrogate models and historical trial data to improve sample efficiency. Meta-Bayesian optimization methods extend this idea by learning priors over surrogate models from related tasks, enabling faster convergence on new optimization problems (Feurer et al., 2015; Perrone et al., 2018). Other approaches, such as ALFA (Baik et al., 2020), adapt hyperparameters dynamically during training using a meta-learner, while SHSR (Borboudakis et al., 2023) prunes unpromising regions of the search space using past AutoML runs. PriorBand (Mallik

et al., 2023) further accelerates HPO by combining expert beliefs with low-fidelity proxy tasks to guide search in deep learning pipelines. These methods illustrate the value of meta-knowledge, but they still assume a fixed model class.

**The CASH Problem.** In practice, algorithms and hyperparameters must be optimized jointly, formalized as the CASH problem (Thornton et al., 2013). A common approach is to treat model choice as a categorical hyperparameter, as in Auto-WEKA (Thornton et al., 2013) and Auto-sklearn (Feurer et al., 2015), but the resulting search space is large and expensive to explore. Bandit-based formulations address this by casting algorithm selection as arms with HPO inside each arm, e.g., MaxUCB (Balef et al., 2025), Rising Bandits (Li et al., 2020), and ER-UCB (Hu et al., 2021). These improve scalability but still depend on extensive search. In contrast, our method tackles CASH directly by generating model and hyperparameter configurations without relying on hierarchical search or bandit-style exploration.

**LLM-Based HPO.** LLMs have recently been applied to hyperparameter optimization, for example through iterative refinement with feedback or by combining with Bayesian optimization (Zhang et al., 2024; Mahammadli & Ertekin, 2025; Liu et al., 2025). While promising, these approaches treat HPO in isolation and require multiple interaction rounds. By contrast, we address the broader CASH problem, producing complete model–hyperparameter configurations in a single inference. AutoML-GPT (Zhang et al., 2023) explores full pipeline automation, including preprocessing, but depends on explicit task similarity matching. Our method is simpler and more practical: we use prior tasks only as in-context examples, letting the LLM adapt implicitly, and we evaluate directly on real-world tabular datasets under standard CASH protocols.

## 3  PROBLEM SETUP

We frame model and hyperparameter selection as a meta-learning problem. Let $\mathcal{P}_{\mathcal{T}}$ denote a distribution over machine learning tasks. For each task $\mathcal{T} \sim \mathcal{P}_{\mathcal{T}}$, we are given a dataset $\mathcal{D}$ and a metadata representation $M$, which summarizes task-level properties such as input dimensionality, sample size, or distributional characteristics. Let $\theta \in \Theta$ denote a model configuration, comprising both the model type and its associated hyperparameters. For a task $\mathcal{T}$, let $L(\theta, \mathcal{T})$ denote the generalization error of configuration $\theta$. The optimal configuration is defined as

$$\theta^* = \arg\min_{\theta \in \Theta} L(\theta, \mathcal{T}).$$

In practice, $\theta^*$ is unknown and must be approximated using train/validation/test splits of the dataset $\mathcal{D}$.

Our objective is to learn a recommendation function $f$ that maps task metadata to a high-performing configuration. Given a new task $\mathcal{T}$, the function receives a metadata instance $M$ along with $k$ support examples $\{(M_1, \theta_1^*), \ldots, (M_k, \theta_k^*)\}$ obtained from past tasks. The function must then predict a configuration $\theta = f(M; M_{1:k}, \theta_{1:k}^*)$ that performs well on $\mathcal{T}$.

In our approach, $f$ is implemented implicitly through in-context learning in a large language model: the LLM receives a prompt containing metadata and possibly prior examples, and outputs a predicted configuration $\theta$. This reduces to a *zero-shot* setting when $k = 0$, where predictions must rely solely on $M$ and prior knowledge encoded in the model. When $k > 0$, the model can perform *meta-informed* prediction by conditioning on past metadata–configuration pairs. To isolate and better understand this behavior, we first study a synthetic classification task where the optimal configuration $\theta^*$ can be computed analytically. We then proceed to evaluate on a suite of real-world tabular benchmark tasks.

## 4  MOTIVATION: SYNTHETIC RIDGE REGRESSION EXPERIMENT

Before evaluating LLM-based model selection on complex benchmarks, we first study a controlled synthetic task: predicting the optimal Ridge regularization parameter $\lambda^*$ for a binary classifier trained on Gaussian data. This setup isolates the meta-learning objective while avoiding confounding factors such as model choice, hyperparameter interactions, and data splits.

**Analytic Test Error.** To evaluate hyperparameter predictions, we require the generalization error of Ridge regression as a function of $\lambda \in \Lambda$. Instead of using costly cross-validation, we leverage a closed-form expression from Random Matrix Theory (Theorem 1 in Appendix A.1), which provides exact test errors and enables precise computation of regret.

*Remark* (Applicability in low dimensions). Although Theorem 1 is formally derived for high-dimensional settings, we verified that it remains accurate even for low-dimensional tasks (e.g., $d = 2$).

**Synthetic Task Setup.** Each task is represented by metadata (class sizes, means, covariances) and the LLM predicts $\lambda^*$ from a fixed logarithmic grid

$$\Lambda = \{10^{-4}, 10^{-3}, \ldots, 10^3\}.$$

For meta-learning evaluation, the LLM is provided with $k$ solved support tasks $(M_i, \lambda_i^*)_{1 \leq i \leq k}$ and a new target task $M$ and must predict the optimal $\lambda$. We vary $k \in \{1, 2, 5, 10, 15, 20, 50, 100\}$ to study how performance improves with more contextual examples.

For each trial, we compute the exact optimal $\lambda^*$ for all tasks using Theorem 1, prompt the LLM with the support tasks and target metadata, and obtain a prediction $\hat{\lambda}$. The predicted value is then rounded to the nearest grid point in $\Lambda$, and performance is measured by regret:

$$\text{Regret} = L(\hat{\lambda}) - L(\lambda^*)$$

Details on task generation and prompt construction are provided in Appendices A.2 and A.3, respectively.

To interpret LLM performance, we consider two baselines:

- **Context-only**: predicts the geometric mean of the support tasks' optimal $\lambda^*$ values, ignoring the target task metadata $M$. This tests whether the LLM simply regresses toward central values from context.

- **Logistic regression**; predicts $\lambda^*$ directly from task metada features. This acts as lightweight supervised meta-learner, simulating the case where cross-task training data is available.

Consistent improvements over both baselines indicates that the LLM leverages task-specific for meaningful adaptation without supervised training.

We evaluate the Qwen 2.5 family (7B, 14B, 32B, 72B) (Qwen et al., 2025), across decoding temperatures $\{0.0, 0.2, 0.4, 0.6, 0.8\}$. Prompt templates are provided in Appendix A.3. To ensure valid outputs, generations are limited to 5 tokens with invalid predictions resampled.

**Results.** To assess the effect of model scale, Figure 2 shows regret as a function of $k$, the number of support tasks. The Qwen2.5 72B model consistently achieves the lowest regret, with its advantage over baselines growing as more context is provided. This indicates that the largest model not only adapts from a few examples, but also continues to benefit from large support sets.

The baselines exhibit distinct limitations. The log-mean method matches LLMs for very small $k$ but quickly saturates at a suboptimal level. Logistic regression improves more gradually and eventually surpasses the log mean, yet it remains far below the 72B model across all $k$.

Smaller LLMs (7B–32B) track the baselines closely and show limited or inconsistent gains as $k$ increases, suggesting weaker in-context adaptation. By contrast, the 72B model demonstrates robust meta-learning: it surpasses both baselines even at large $k$ and continues to improve steadily with more support tasks.

Finally, we verified that the decoding temperature (0.0–0.8) has no measurable effect on regret across any model, confirming that our results are robust to this choice (see Appendix A.4 for detailed plots). Overall, these findings suggest that sufficiently large LLMs can learn to generalize hyperparameter selection strategies from sparse supervision, without parameter updates.

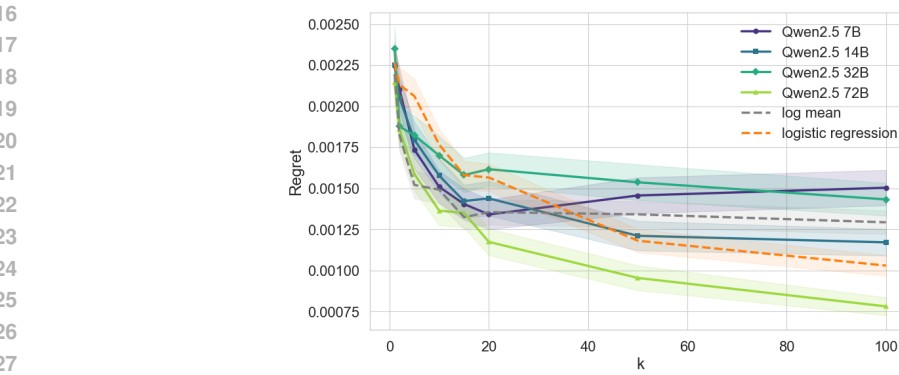

Figure 2: Regret vs. number of support tasks $k$, averaged across decoding temperatures. The dashed line represents a static geometric-mean baseline. Shaded regions denote 90% confidence intervals: for model predictions, intervals are computed from the standard error over 5000 trials (1000 per temperature); for the baselines, intervals reflect 1000 trials. The 72B model is the only model to consistently outperform the baselines as $k$ increases, indicating scale-dependent emergence of in-context meta-learning.

## 5 METHODOLOGY AND EXPERIMENTS

We now describe our general evaluation framework and present empirical results on real-world tabular regression and classification benchmarks. The methodology extends the setup from Section 3, and the experiments test whether the in-context meta-learning behaviors observed in the synthetic ridge regression setting also emerge in practical classification and regression tasks.

### 5.1 METHODOLOGY

As formalized in Section 3, each task $\mathcal{T}_i$ is represented by a metadata block $M_i$, and the goal is to predict a configuration $\theta_i$ consisting of a set of models and their hyperparameters. In our setting, this set is intended to form an ensemble: the LLM proposes multiple candidate models whose predictions are later combined through the ensembling pipeline. We implement this mapping $f : M_i \mapsto \theta_i$ through in-context learning in a large language model.

```
# Metadata for kaggle_abalone

## prediction_type
regression
## score_name
rmsle
## n_train: 90615 n_test: 60411
## features
total: 9
numeric: 8 categorical: 1
## missing_data
has_missing: False
## target_values min:
1 max: 29 mean: 9.697 std: 3.176
```

**Task metadata.** We summarize each dataset using a fixed Markdown-style template designed for compactness and interpretability. The metadata captures prediction type, evaluation metric, sample sizes, feature composition (numeric vs. categorical), missingness indicators, and target statistics. Rather than enumerating every feature, which would make prompts impractically long for high-dimensional datasets, the template records only aggregated statistics (e.g., counts of feature types, summary ranges). A simplified example for the `abalone` challenge is shown on the left, and the full schema is provided in Appendix C.1.

We compared Markdown and JSON encodings, finding that Markdown reduced token length by roughly 30% without degrading recommendation quality. This efficiency allows more support examples to be included in-context while keeping prompts short and interpretable.

**Prompting strategies.** We evaluate two prompting modes:

- **Zero-Shot:** the LLM receives only the target metadata $M_j$, relying solely on pretrained knowledge.

- **Meta-Informed:** the LLM additionally observes a set of solved support tasks $\{(M_i, \theta_i^*)\}_{i=1}^k$, all drawn from the same prediction type (classification or regression). In this setting, the model is explicitly asked to identify similarities between tasks before recommending $\theta_j$.

In practice, the **Meta-Informed** strategy assumes access to previous tasks along with high-performing configurations. For this study, we obtained such configurations by running extensive hyperparameter search with HEBO (Cowen-Rivers et al., 2022) on a set of tabular regression and classification tasks. To maximize performance, ensembles (or blends) were built from the resulting models. We refer to the models with the highest contributions to these ensembles as **Context Blends**, and use them as the source of support examples passed to the prompt.

**Configuration schema and hyperparameter grids.** The LLM is instructed to output a JSON object describing an ensemble of 10 models. For each supported family (CatBoost (Prokhorenkova et al., 2018), LightGBM, XGBoost (Chen & Guestrin, 2016), and scikit-learn MLP (Pedregosa et al., 2011)), we provide the model name, a list of valid hyperparameters, and a discrete grid of admissible values. This grid is included directly in the prompt, ensuring that the model generates configurations from a well-defined search space rather than free-form values. An excerpt of the schema is shown below (see Appendix F for full hyperparamater grids):

```
{
  "models": {
    "catboost": {
      "columns": ["bootstrap_type", "border_count", "grow_policy", ...],
      "values": []
    },
    "lgbm": {
      "columns": ["boosting_type", "colsample_bynode", "drop_rate", ...],
      "values": []
    },
    ...
  }
}
```

**Reasoning and output validation.** We use the DeepSeek-R1 reasoning model (DeepSeek-AI et al., 2025), which naturally produces explanations of its choices. The LLM configuration is described in Appendix D. Invalid generations are rare, but we apply lightweight post-processing when they occur. If the LLM outputs a numeric value that falls outside the predefined hyperparameter grid, we project it to the nearest valid grid point. For non-numeric fields (e.g., categorical options) that cannot be matched, we discard the configuration and resample a fresh output. Likewise, if the JSON structure itself is malformed, the entire configuration is rejected and regenerated. Each run uses a different set of support examples, ensuring robustness to contextual variation.

**Prompt length and overhead.** Prompt lengths remain modest: Zero-Shot prompts contain only one metadata block, while Meta-Informed prompts add up to $k$ support examples. In practice, the LLM forward pass incurs negligible cost compared to training the resulting models, making the overhead essentially free relative to model training.

**Ensembling pipeline.** Each LLM call outputs 10 configurations, which we treat as candidate base models. We train these with cross-validation bagging and then combine their predictions using feedforward greedy blending (Caruana et al., 2004). This procedure is applied consistently to LLM-based and baseline methods, providing a fair comparison and reflecting common ML ensembling practice.

## 5.2 DATASETS

We evaluate our method on 22 Kaggle tabular challenges spanning both regression and classification. The benchmark covers a mix of "playground" competitions (synthetic or repurposed datasets) and "featured" challenges (industrial or scientific applications), providing a broad spectrum of problem settings. Kaggle tasks are particularly suitable for this study because they provide standardized train/test splits, diverse evaluation metrics, and well-documented leaderboards, which together ensure reproducibility and facilitate comparison with baselines.

Prediction types range from regression to binary and multi-class classification, with metrics including error-based losses (RMSE, MAE, RMSLE), probabilistic measures (AUC, log-loss, NLL), and discrete scores (accuracy, $F_1$). Dataset scales vary widely from fewer than 2,000 training points (`horses`) to several hundred thousand (`media`, `insurance`), while feature dimensionality ranges from fewer than 10 (`abalone`) to over a thousand (`molecules`). This diversity ensures coverage of small vs. large data regimes, low- vs. high-dimensional settings, and synthetic vs. real-world tasks. The full dataset list with detailed statistics is provided in Table 2 in the Appendix.

### 5.3 BASELINES

We compare LLM-based recommendations against four baselines representing different strategies for the CASH problem (full details in Appendix G): **Context-Random** (uniformly samples model–hyperparameter configurations from the same reference pool as the one passed to the LLM), **Random-Hyperopt** (at each step, uniformly samples a model family and then applies a hyperparameter optimizer within that family), **LGBM-Hyperopt** (optimizer restricted to LightGBM, capturing the strength of a single tuned family), and **MaxUCB-Hyperopt** (treats each family as a bandit arm, selecting the one with the highest upper-confidence bound before a single optimization step (Balef et al., 2025)). **Context Blends** consist of ensembles obtained from an extensive hyperparameter search. They provide upper-bound baselines: they achieve high performance through extensive search, and thus set the performance we seek to approach under a much more limited budget. All **-Hyperopt** baselines use HEBO (Cowen-Rivers et al., 2022), chosen for its strong and consistent performance across diverse tasks (Kegl, 2023)[1].

### 5.4 EVALUATION METRIC

We assess blend quality using the private leaderboard percentile rank ($p_{\text{rank}}$), which measures the percentage of submissions beaten by a given configuration on Kaggle's hidden test set. A value of $p_{\text{rank}} = 100$ indicates the top submission on the leaderboard, while $p_{\text{rank}} = 0$ corresponds to the lowest. This metric is scale-invariant across datasets with different evaluation metrics and directly reflects the competitive standard of Kaggle challenges. We report mean $p_{\text{rank}}$ across tasks, with uncertainty estimated from the standard error over random seeds.

### 5.5 PERFORMANCE COMPARISON

We compare LLM-generated ensembles against the baselines introduced in Section 5.3, using the private leaderboard percentile rank ($p_{\text{rank}}$; higher is better) as our evaluation metric. For fairness, all methods are restricted to training exactly 10 models on each dataset. This provides a comparable runtime budget across methods, since model training is the dominant cost irrespective of how configurations are proposed.

**Results.** Blend quality is measured using the private leaderboard percentile rank (p-rank; higher is better) after training on the Kaggle datasets. Figure 3 summarizes the average performance across 22 datasets. **Meta-Informed** achieves the strongest LLM-driven performance (72.7), surpassing both **Zero-Shot** (70.4) and **Context-Random** (70.0), while clearly outperforming Hyperopt based baselines including the best one **Random-Hyperopt** (65.7). Although the AutoML-derived **Context Blends** remains higher (77.7), this performance is achieved at

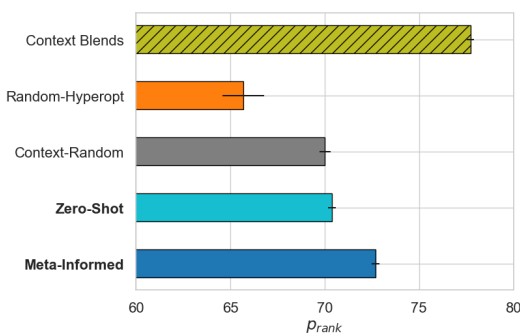

Figure 3: Comparison of prompting strategies and baselines in terms of $p_{\text{rank}}$. The **Context Blends** produced by AutoML performance for each challenge are shown as a reference. Error bars indicate 90% confidence intervals of the mean across 8 random seeds per dataset.

---

[1]HEBO begins with random search, using $1 +$ (dimension of the hyperparameter space) evaluations, before switching to Bayesian optimization.

the cost of a much more expensive procedure, whereas our strategies rely on training only 10 models. Importantly, the significant improvement of **Meta-Informed** over **Context-Random** indicates that the LLM is not merely sampling from the metadata, but is leveraging past tasks' information in a way that reflects genuine adaptation. Finally, across most datasets (Table 1), LLM-based methods exhibit lower uncertainty than Hyperopt baselines, indicating more stable performance. A more detailed analysis of per-dataset patterns is provided in Appendix B.2.

Table 1: Kaggle p-rank results across all challenges. Uncertainty is reported as $\pm$ values, representing the 90% confidence interval based on the standard error across 8 random seeds.

| Kaggle Challenge | Meta-Informed | Zero-Shot | Context-Random | Random-Hyperopt | MaxUCB-Hyperopt | LGBM-Hyperopt |
|---|---|---|---|---|---|---|
| abalone | $85.73 \pm 3.3$ | $74.67 \pm 4.6$ | $\mathbf{87.87 \pm 2.3}$ | $58.95 \pm 4.6$ | $56.53 \pm 9.0$ | $64.21 \pm 11.3$ |
| allstate | $\mathbf{69.92 \pm 2.3}$ | $61.66 \pm 2.9$ | $65.41 \pm 5.0$ | $50.05 \pm 2.4$ | $56.25 \pm 2.7$ | $51.0 \pm 2.7$ |
| attrition | $59.51 \pm 1.7$ | $\mathbf{61.12 \pm 1.8}$ | $57.31 \pm 2.3$ | $59.36 \pm 3.3$ | $58.69 \pm 2.6$ | $48.21 \pm 5.0$ |
| blueberry | $\mathbf{81.16 \pm 2.4}$ | $79.86 \pm 1.7$ | $78.96 \pm 3.8$ | $70.77 \pm 5.3$ | $62.9 \pm 7.1$ | $65.87 \pm 7.7$ |
| churn | $70.35 \pm 0.9$ | $68.73 \pm 0.9$ | $68.71 \pm 3.0$ | $65.07 \pm 4.0$ | $62.98 \pm 6.0$ | $\mathbf{70.64 \pm 1.0}$ |
| cirrhosis | $70.58 \pm 3.6$ | $69.09 \pm 1.4$ | $\mathbf{73.06 \pm 1.8}$ | $64.61 \pm 4.6$ | $66.96 \pm 1.9$ | $70.17 \pm 2.0$ |
| concrete strength | $74.34 \pm 17.9$ | $74.19 \pm 6.8$ | $59.37 \pm 16.1$ | $\mathbf{88.81 \pm 5.4}$ | $75.46 \pm 13.8$ | $83.21 \pm 9.3$ |
| covertype | $\mathbf{67.78 \pm 4.0}$ | $58.35 \pm 7.6$ | $60.05 \pm 10.3$ | $56.75 \pm 11.0$ | $53.75 \pm 6.2$ | $32.0 \pm 3.4$ |
| crab age | $\mathbf{68.87 \pm 0.7}$ | $68.81 \pm 0.6$ | $67.67 \pm 1.2$ | $61.84 \pm 2.3$ | $59.53 \pm 3.2$ | $63.84 \pm 1.8$ |
| credit fusion | $96.61 \pm 1.0$ | $96.71 \pm 1.1$ | $90.91 \pm 1.7$ | $96.35 \pm 0.9$ | $94.12 \pm 1.8$ | $\mathbf{96.75 \pm 1.5}$ |
| failure | $41.12 \pm 1.5$ | $43.52 \pm 1.7$ | $41.25 \pm 0.8$ | $43.7 \pm 2.6$ | $47.15 \pm 5.0$ | $\mathbf{48.15 \pm 7.0}$ |
| heat flux fi | $\mathbf{93.4 \pm 5.0}$ | $90.7 \pm 4.3$ | $83.65 \pm 8.6$ | $69.07 \pm 6.6$ | $47.37 \pm 11.3$ | $36.22 \pm 17.1$ |
| horses | $82.39 \pm 7.7$ | $\mathbf{82.78 \pm 5.6}$ | $75.31 \pm 10.6$ | $81.15 \pm 6.2$ | $72.7 \pm 9.2$ | $79.75 \pm 5.7$ |
| housing california | $\mathbf{62.53 \pm 0.6}$ | $54.84 \pm 2.4$ | $60.07 \pm 2.0$ | $46.9 \pm 6.8$ | $42.15 \pm 8.2$ | $52.71 \pm 3.9$ |
| influencers | $76.84 \pm 7.4$ | $\mathbf{83.55 \pm 1.4}$ | $80.52 \pm 2.8$ | $82.95 \pm 2.7$ | $82.03 \pm 3.0$ | $87.45 \pm 1.9$ |
| insurance | $\mathbf{74.68 \pm 2.4}$ | $68.16 \pm 1.8$ | $67.9 \pm 2.1$ | $62.53 \pm 5.9$ | $66.76 \pm 4.2$ | $64.6 \pm 3.4$ |
| loan approval | $71.58 \pm 2.6$ | $63.29 \pm 5.5$ | $66.84 \pm 5.4$ | $62.64 \pm 6.9$ | $60.81 \pm 4.8$ | $\mathbf{74.43 \pm 0.9}$ |
| media | $\mathbf{62.95 \pm 1.4}$ | $57.52 \pm 2.0$ | $61.81 \pm 2.5$ | $49.5 \pm 7.5$ | $47.87 \pm 5.6$ | $26.07 \pm 2.8$ |
| mental health | $\mathbf{92.99 \pm 3.0}$ | $79.77 \pm 10.2$ | $89.69 \pm 5.2$ | $75.34 \pm 9.5$ | $73.39 \pm 9.3$ | $80.11 \pm 7.7$ |
| mercedes | $17.81 \pm 2.8$ | $36.44 \pm 7.8$ | $35.26 \pm 10.6$ | $36.57 \pm 8.6$ | $\mathbf{38.94 \pm 4.7}$ | $25.42 \pm 2.0$ |
| molecules | $\mathbf{97.52 \pm 1.5}$ | $96.34 \pm 1.6$ | $96.32 \pm 3.3$ | $96.33 \pm 2.6$ | $94.84 \pm 1.9$ | $78.02 \pm 12.6$ |
| unknown a | $\mathbf{80.56 \pm 0.8}$ | $78.6 \pm 0.8$ | $72.59 \pm 2.4$ | $66.17 \pm 2.5$ | $61.75 \pm 6.0$ | $61.41 \pm 5.5$ |
| **Mean** | $\mathbf{72.69 \pm 0.2}$ | $70.39 \pm 0.2$ | $70.02 \pm 0.3$ | $65.7 \pm 1.1$ | $62.86 \pm 1.2$ | $61.8 \pm 1.1$ |

## 5.6 PERFORMANCE EFFICIENCY

To complement performance ranking, we also evaluate efficiency relative to standard hyperparameter optimization. For this comparison, we focus on a subset of six datasets: `abalone`, `blueberry`, `covertype`, `heat flux fi`, `horses`, and `media`.

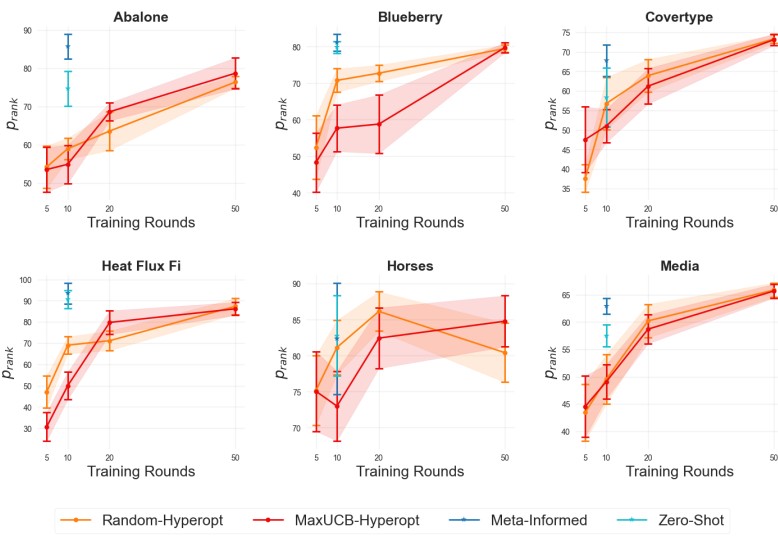

Figure 4: $p_{\text{rank}}$ over training *rounds* for **Random-Hyperopt**, **MaxUCB-Hyperopt**, **Meta-Informed**, and **Zero-Shot** across the six selected datasets. Error bars indicate 90% confidence intervals using standard error across 8 seeds.

We define one *round* as training a single model configuration followed by its integration into the blending pipeline, ensuring all methods incur the same per-round cost. The LLM based methods (**Zero-Shot** and **Meta-Informed**) produce exactly ten configurations in a single forward pass and thus correspond to a budget of 10 rounds. By contrast, **Random-Hyperopt** and **MaxUCB-Hyperopt** can continue to propose new configurations sequentially and we evaluate their performance after 5, 10, 20 and 50 rounds.

**Results.**  On five of these six datasets, the LLM based methods match or exceed performance of Hyperopt ones within the same budget of ten training rounds, while Hyperopt methods seems to require substantially more rounds to achieve similar performance (Figure 4). This highlights an efficiency advantage when measured on a per-round basis: LLM-based methods deliver high-quality configurations immediately, whereas Hyperopt ones improve only gradually through extended exploration. In practice, this advantage could be even more pronounced since LLMs produce all of their candidates in a single inference step. This means that the full set of configurations is available upfront and can be trained in parallel, while Hyperopt methods must generate candidates one at a time, limiting opportunities for parallelization and slowing down the overall search process.

### 5.7 INTERPRETABILITY

Another advantage of LLM-based methods is interpretability. Unlike conventional hyperparameter optimization, which produces configurations without explanation, the LLM generates structured outputs accompanied by reasoning traces. These traces highlight how the model can relate task metadata to past examples when proposing new model–hyperparameter ensembles. For example, the LLM often explains its choices by linking dataset properties to its choices such as favoring CatBoost on feature sets dominated by categorical variables, or suggesting deeper trees when the regression task involves many numeric features. Appendix E presents selected reasoning traces that illustrate how the model draws on prior tasks and/or its internal knowledge to guide model and hyperparameter recommendations.

## 6 DISCUSSION

While our results establish the competitiveness of LLM-based CASH strategies, they also outline challenges that remain to be addressed. As detailed in Appendix B.2, performance on small datasets or those with extreme feature-to-sample ratios is less consistent, pointing to a dependence on richer metadata for reliable adaptation. This suggests that characterizing the conditions under which LLMs succeed or fail will be an important direction for future work. The methods proved stable to shuffling the order of items within the prompt (Appendix H), suggesting that performance is not strongly tied to positional artifacts. Finally, our study restricted evaluation to four model families for tractability, but extending coverage to a broader set of models and hyperparameters will be essential for assessing generality and exploring the full potential of LLM-based CASH.

## 7 CONCLUSION

Our experiments show that large language models can exploit metadata from support tasks to recommend models and hyperparameters competitively without iterative search. They also provide strong task-dependent defaults, offering practitioners a practical starting point without extensive tuning. These results demonstrate the viability of LLMs as in-context meta-learners for the CASH problem and highlight their potential as an efficient complement to conventional AutoML pipelines.

REPRODUCIBILITY STATEMENT

Details of the synthetic experiment protocol are given in Appendix A. All datasets used in this work are publicly available Kaggle challenges, with detailed statistics and links provided in Appendix B. For the Kaggle experiments, metadata schemas, prompt templates, base model grids, and LLM configurations are specified in Appendices C, F, and D. Baseline implementations follow published protocols, with hyperparameter search details described in Section 5.3 and Appendix G. To further ensure reproducibility, we will release the code to reproduce the experiments once the paper has been accepted at [Link hidden for review].

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

# Appendix

# A  SYNTHETIC RIDGE EXPERIMENT

## A.1  CLOSED-FORM TEST ERROR

**Notation.**  Throughout the appendix, bold uppercase letters (e.g. $\mathbf{A}$) denote matrices, bold lowercase letters (e.g. $\mathbf{x}$) denote vectors, and plain lowercase letters (e.g. $x$) denote scalars. We use $\|\mathbf{x}\|_2$ for the Euclidean norm of a vector, $\|\mathbf{A}\|$ for the spectral (operator) norm of a matrix, and $\|\mathbf{A}\|_F$ for its Frobenius norm. For two sequences of real numbers $u_n$ and $v_n$, the notation $u_n = O(v_n)$ indicates that $|u_n/v_n|$ remains bounded (as $n \to \infty$), typically with high probability. Expectation is denoted by $\mathbb{E}[\cdot]$.

**Setup.**  We consider a *binary classification* problem in $d$-dimensional space. Fix a dimension $d \in \mathbb{N}$. We observe a labeled training sample
$$\left\{ (\mathbf{x}_i, y_i) \right\}_{i=1}^n,$$
where for each $i = 1, \dots, n$:

- $\mathbf{x}_i \in \mathbb{R}^d$ is the $d$-dimensional feature vector,
- $y_i \in \{+1, -1\}$ is the corresponding class label.

We assume that the data come from a mixture of two Gaussian classes:
$$\mathbf{x} \,\big|\, y = +1 \sim \mathcal{N}(\mu_1, \mathbf{\Sigma}_1), \qquad \mathbf{x} \,\big|\, y = -1 \sim \mathcal{N}(\mu_2, \mathbf{\Sigma}_2),$$
Let $n_k$ be the number of training samples from class $k \in \{1, 2\}$, $n = n_1 + n_2$. Define class proportions $c_k := n_k/n$. Denote
$$\mathbf{C}_k := \mathbf{\Sigma}_k + \mu_k \mu_k^\top \in \mathbb{R}^{d \times d}, \qquad k = 1, 2.$$

**Ridge regression classifier.**  Given the training set $\{(\mathbf{x}_i, y_i)\}_{i=1}^n$ with $\mathbf{x}_i \in \mathbb{R}^d$ and $y_i \in \{+1, -1\}$, we train a *ridge regression classifier* (least-squares with $\ell_2$ penalty). Specifically, for a regularization parameter $\lambda > 0$ we solve
$$\widehat{\mathbf{w}}(\lambda) \;=\; \arg\min_{\mathbf{w} \in \mathbb{R}^d} \; \frac{1}{n} \sum_{i=1}^n \big(y_i - \mathbf{w}^\top \mathbf{x}_i\big)^2 + \lambda \|\mathbf{w}\|_2^2. \tag{1}$$

This is the standard ridge regression problem. Its closed-form solution is
$$\widehat{\mathbf{w}}(\lambda) \;=\; \big(\mathbf{X}^\top \mathbf{X}/n + \lambda \mathbf{I}_d\big)^{-1} \mathbf{X}^\top \mathbf{y}/n, \tag{2}$$
where $\mathbf{X} \in \mathbb{R}^{n \times d}$ is the data matrix with rows $\mathbf{x}_i^\top$ and $\mathbf{y} = (y_1, \dots, y_n)^\top$ the label vector.

Given a new test point $\mathbf{x} \in \mathbb{R}^d$, the classifier computes the *score*
$$s(\mathbf{x}) \;=\; \widehat{\mathbf{w}}(\lambda)^\top \mathbf{x}, \tag{3}$$
which is then compared to a decision threshold (e.g. zero or an optimally chosen $\eta^\star$) to produce a predicted label.

All formulas below are deterministic equivalents / asymptotic formulas obtained by the standard Gaussian and random-matrix approximations used to derive fixed point equations.

*Assumption* 1. (Regularity / high-dimensional regime) The feature dimension $d$ and the sample sizes $n_k$ grow so that: $d, n_1, n_2 \to \infty$ with $d/n \to \gamma \in (0, \infty)$ and $c_k = n_k/n \to \bar{c}_k \in (0, \infty)$. The family of pair $(\mathbf{\Sigma}_1, \mathbf{\Sigma}_2)$ is uniformly bounded in operator norm and their empirical spectral distributions admit limits.

**Auxiliary fixed point definitions.**  For a given $\lambda > 0$ we seek $\delta = (\delta_1, \delta_2) \in \mathbb{R}^2$ and a matrix $\bar{\mathbf{Q}}(\lambda) \in \mathbb{R}^{d \times d}$ defined implicitly by the equations
$$\bar{\mathbf{Q}}(\lambda) := \left( \sum_{k=1}^2 \frac{c_k}{1 + \delta_k} \, \mathbf{C}_k + \lambda \mathbf{I}_d \right)^{-1}, \tag{4}$$
$$\delta_k = \frac{1}{n} \operatorname{tr}\big(\mathbf{C}_k \, \bar{\mathbf{Q}}(\lambda)\big), \qquad k = 1, 2. \tag{5}$$

The existence and uniqueness of a positive solution follow under the above regularity conditions; numerically $\delta$ is found by simple fixed-point iteration.

Define the diagonal scaling matrix and the (scaled) mean matrix

$$\mathcal{D}_\delta := \mathrm{diag}\big(\tfrac{1}{1+\delta_1}, \tfrac{1}{1+\delta_2}\big), \qquad \mathbf{M}_\delta := \mathcal{D}_\delta \begin{bmatrix} \mu_1^\top \\ \mu_2^\top \end{bmatrix} \in \mathbb{R}^{2\times d}.$$

We also define two $d \times d$ matrices $\mathbf{K}_1, \mathbf{K}_2$ and two 2-vectors $d^{(1)}, d^{(2)}$ through the linear algebraic operations below:

$$\mathbf{V} := \frac{1}{n} \begin{bmatrix} \mathrm{tr}(\mathbf{C}_1\bar{\mathbf{Q}}\mathbf{C}_1\bar{\mathbf{Q}}) & \mathrm{tr}(\mathbf{C}_1\bar{\mathbf{Q}}\mathbf{C}_2\bar{\mathbf{Q}}) \\ \mathrm{tr}(\mathbf{C}_2\bar{\mathbf{Q}}\mathbf{C}_1\bar{\mathbf{Q}}) & \mathrm{tr}(\mathbf{C}_2\bar{\mathbf{Q}}\mathbf{C}_2\bar{\mathbf{Q}}) \end{bmatrix}, \qquad \mathbf{A} := \mathrm{diag}\Big(\frac{c_1}{(1+\delta_1)^2}, \frac{c_2}{(1+\delta_2)^2}\Big),$$

$$\mathbf{t}^{(j)} := \begin{bmatrix} \frac{1}{n}\,\mathrm{tr}(\mathbf{C}_1\bar{\mathbf{Q}}\boldsymbol{\Sigma}_j\bar{\mathbf{Q}}) \\ \frac{1}{n}\,\mathrm{tr}(\mathbf{C}_2\bar{\mathbf{Q}}\boldsymbol{\Sigma}_j\bar{\mathbf{Q}}) \end{bmatrix}, \qquad j = 1, 2,$$

$$\mathbf{d}^{(j)} := (\mathbf{I}_2 - \mathbf{VA})^{-1}\mathbf{t}^{(j)}, \qquad j = 1, 2,$$

and then

$$\mathbf{K}_j := \bar{\mathbf{Q}}\,\boldsymbol{\Sigma}_j\,\bar{\mathbf{Q}} \;+\; \frac{c_2 d_1^{(j)}}{(1+\delta_1)^2}\bar{\mathbf{Q}}\mathbf{C}_1\bar{\mathbf{Q}} \;+\; \frac{c_1 d_2^{(j)}}{(1+\delta_2)^2}\bar{\mathbf{Q}}\mathbf{C}_2\bar{\mathbf{Q}}, \qquad j = 1, 2.$$

Define the asymptotic (deterministic) class scores' means and variances as follows.

Let $\mathbf{y}$ be the vector of training labels with entries $+1$ for class 1 samples and $-1$ for class 2 samples, and write $\mathbf{J} \in \mathbb{R}^{n\times 2}$ for the class indicator matrix with columns equal to the indicators of class membership. Then the limiting (deterministic) score means are

$$m_k = \frac{1}{n}\,\mathbf{y}^\top \mathbf{J}\,\mathbf{M}_\delta\,\bar{\mathbf{Q}}\,\mu_k, \qquad k = 1, 2,$$

and the limiting score variances are

$$v_k = \frac{1}{n^2}\Big(\mathbf{y}^\top \mathbf{V}^{(k)}\mathbf{y} + \mathbf{y}^\top \mathbf{JM}_\delta\mathbf{K}_k\mathbf{M}_\delta^\top\mathbf{J}^\top\mathbf{y} - 2\,\mathbf{y}^\top \mathbf{JM}_{\delta,\Delta}^{(k)}\bar{\mathbf{Q}}\mathbf{M}_\delta^\top\mathbf{J}^\top\mathbf{y}\Big),$$

where $\mathbf{V}^{(k)}$ is the diagonal matrix whose entries are the per-sample variances built from $\mathrm{tr}(\boldsymbol{\Sigma}_i\mathbf{K}_k)/(1+\delta_i)^2$, and $\mathbf{M}_{\delta,\Delta}^{(k)}$ is the matrix built from the traces $\mathrm{tr}(\boldsymbol{\Sigma}_i\mathbf{K}_k)$.

**Theorem 1** (Asymptotic Gaussianity and deterministic test error). *Under the assumptions above, for any fixed regularization $\lambda > 0$ the distribution of the ridge score $s(\mathbf{x}) = \widehat{\mathbf{w}}(\lambda)^\top\mathbf{x}$ conditional on $\mathbf{x}$ belonging to class $k$ converges in distribution to a Gaussian with mean $m_k$ and variance $v_k$ as $d, n \to \infty$. That is,*

$$s(\mathbf{x}) \mid (\mathbf{x} \sim \text{class } k) \xrightarrow{d} \mathcal{N}(m_k, v_k), \qquad k = 1, 2,$$

*where $m_k, v_k$ are given by the deterministic formulas above (they are computed from the unique solution of the fixed point system equation 4–equation 5 together with the algebraic definitions of $\mathbf{K}_k$).*

*Consequently, the asymptotic test error (balanced between the two classes) for the optimal threshold $\eta^\star$ that minimizes the misclassification probability equals*

$$\mathcal{E}(\lambda) = \tfrac{1}{2}\,\Phi\Big(\frac{\eta^\star - m_{\max}}{\sqrt{v_{\max}}}\Big) + \tfrac{1}{2}\Big(1 - \Phi\Big(\frac{\eta^\star - m_{\min}}{\sqrt{v_{\min}}}\Big)\Big),$$

*where $m_{\max} = \max\{m_1, m_2\}$, $m_{\min} = \min\{m_1, m_2\}$, and $v_{\max}, v_{\min}$ are the variances corresponding to those means. The optimal threshold $\eta^\star$ is the solution of*

$$\frac{1}{\sqrt{v_1}}\,(\eta - m_1) = \pm\frac{1}{\sqrt{v_2}}\,(\eta - m_2),$$

*Proof sketch.* The proof is a combination of two standard ingredients:

1. *Deterministic equivalents / resolvent fixed point.* Using standard random-matrix techniques (resolvent identities and deterministic equivalents for sample covariance resolvents) (Couillet & Debbah, 2011, Chapter 6), one shows that the random matrix inverse that appears in the ridge formula concentrates around the deterministic matrix $\bar{\mathbf{Q}}(\lambda)$ defined in equation 4 and that the scalar traces $(1/n)\operatorname{tr}(\mathbf{C}_k\bar{\mathbf{Q}})$ converge to the solution $\delta_k$ of equation 5. This gives the first-order deterministic equivalents used to compute $m_k$.

2. *Gaussian fluctuation / CLT.* After centering by the deterministic mean, the score is a linear or quadratic form of Gaussian vectors; a multivariate CLT (together with second-order deterministic equivalents captured by $\mathbf{K}_k$ and the $\mathbf{d}^{(j)}$ corrections) yields asymptotic Gaussianity Tiomoko et al. (2020) with variance given by the deterministic formula $v_k$.

□

## A.2 TASK GENERATION PROCEDURE

We generate synthetic tasks $\mathcal{T}$ as binary Gaussian classification problems of dimension $d = 2$.

$$\mathcal{T} = \big(n_1, n_2, \mu_1, \mu_2, \alpha_1, \alpha_2\big),$$

with the following components:

- $n_1, n_2$: sample counts for classes 1 and 2, drawn uniformly at random from $\{10, \ldots, 500\}$.

- $\mu_1, \mu_2$: mean vectors of the two classes. We fix

$$\mu_1 = (1, 1, \ldots, 1) \in \mathbb{R}^d,$$

and define

$$\mu_2 = -\varepsilon \cdot (1, 1, \ldots, 1),$$

where $\varepsilon \in [0, 2]^2$ is sampled i.i.d. from the uniform distribution and rounded to two decimal places.

- $\alpha_1, \alpha_2$: AR(1) Toeplitz correlation coefficients, drawn uniformly from $[0, 0.9]$ (rounded to two decimal places). These define the covariance matrices

$$\Sigma_{ij}^{(c)} = \alpha_c^{|i-j|}, \qquad c \in \{1, 2\}.$$

Hence, each task $\mathcal{T}$ specifies two Gaussian distributions

$$X \mid Y = c \ \sim \ \mathcal{N}\big(\mu_c, \Sigma^{(c)}\big), \quad c \in \{1, 2\},$$

together with their respective sample sizes $n_c$.

Because the class means, covariances, and sample sizes are randomized across tasks, the resulting problems differ in signal-to-noise ratio and feature correlations. Consequently, the optimal ridge regularization parameter $\lambda^\star$ varies substantially.

## A.3 PROMPTS

We query the LLM to select an optimal ridge penalty $\lambda$ from a fixed grid given JSON task metadata. Two prompt variants are used: (i) a *zero-shot* prompt with no past tasks, and (ii) a *meta-informed* prompt with a list of past tasks annotated with their optimal $\lambda^\star$.

**Zero-Shot Prompt**

You are a statistics assistant. Your task is to inspect a Gaussian classification problem that will be solved with ridge regression and then pick the optimal ridge regularisation constant lambda for this problem (task_id: 0). The task is a two-class Gaussian problem with:
• n1, n2 : sample counts for classes 1 and 2;
• mu1, mu2 : mean vectors of the two classes;
• alpha1, alpha2 : AR(1) Toeplitz correlation coefficients defining each class's covariance Sigma_ij = alpha $\hat{}|i - j|$.
Choose lambda only from the common grid provided below.

# Common lambda grid (shared by every task):

{{LAMBDA_GRID_JSON}}

# Task (predict lambda_star). Pick \*\*exactly one\*\* lambda from the common grid above that minimises test error for this task. Output just that number, no extra text.

{{NEW_TASK_JSON}}

**Meta-Informed Prompt**

You are a statistics assistant.Your task is to inspect several past Gaussian classification problems that were solved with ridge regression and then pick the optimal ridge regularisation constant lambda for ONE new problem (task_id: NEW). The task is a two-class Gaussian problem with:
• n1, n2 : sample counts for classes 1 and 2;
• mu1, mu2 : mean vectors of the two classes;
• alpha1, alpha2 : AR(1) Toeplitz correlation coefficients defining each class's covariance Sigma_ij = alpha $\hat{}|i - j|$.
Choose lambda only from the common grid provided below.

# Common lambda grid (shared by every task):

{{LAMBDA_GRID_JSON}}

# Past tasks with known optimal lambda_star:

{{PAST_TASKS_JSON}}

# Task (predict lambda_star). Pick \*\*exactly one\*\* lambda from the common grid above that minimises test error for this task. Output just that number, no extra text.

{{NEW_TASK_JSON}}

## A.4 EFFECT OF DECODING TEMPERATURE

We examined the impact of decoding temperature on regret across all LLMs. Temperatures $T \in \{0.0, 0.2, 0.4, 0.6, 0.8\}$ were tested using the same protocol as in the main experiments. Figure 5 reports the results.

Across all models, we observe that decoding temperature has only a marginal effect on regret with the confidence intervals overlapping substantially. This indicates that regret is largely insensitive to sampling temperature, and thus our main results are robust to this choice.

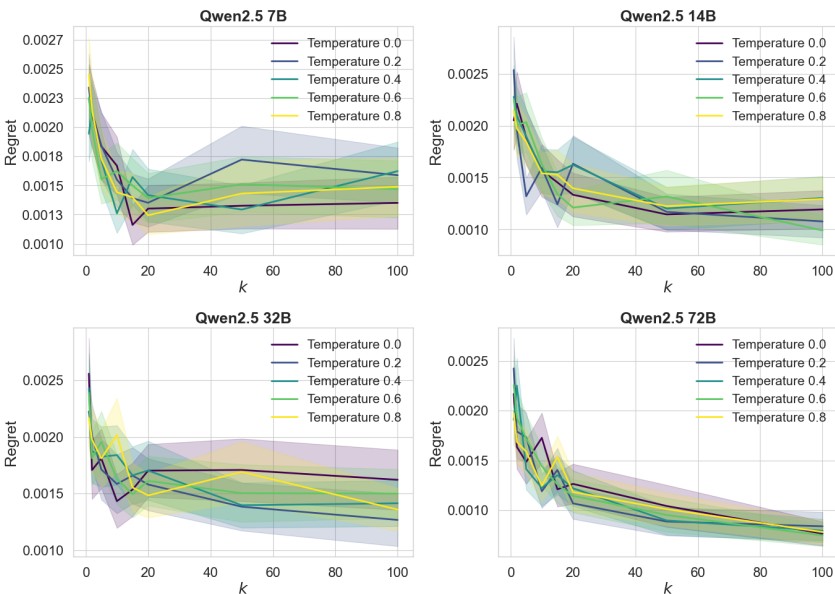

Figure 5: Regret vs. number of support tasks $k$ for Qwen 2.5 models at five decoding temperatures (T=0.0 to 0.8). Shaded regions denote 90% confidence interval based on standard error across 1000 trials. Only the 72B model shows consistent improvement with increasing k, with minimal effect of temperature across all models.

# B  KAGGLE BENCHMARK DETAILS

## B.1  KAGGLE CHALLENGES

Table 2 summarizes the statistics of the tabular challenges used in this paper, highlighting a wide range of problem types, metrics, and data sizes.

| Kaggle challenge | type | year | pred type | metric | # team | # train | # test | # feat | # cat | # num | # cls | # miss |
|---|---|---|---|---|---|---|---|---|---|---|---|---|
| abalone | play | 2024 | reg | rmsle | 2606 | 90615 | 60411 | 8 | 1 | 7 | | 0 |
| allstate | feat | 2016 | reg | mae | 3045 | 188318 | 125546 | 130 | 116 | 14 | | 0 |
| attrition | play | 2023 | bin | auc | 665 | 1677 | 1119 | 33 | 8 | 25 | 2 | 0 |
| blueberry | play | 2023 | reg | mae | 1875 | 15289 | 10194 | 16 | 0 | 16 | | 0 |
| churn | play | 2024 | bin | auc | 3632 | 165034 | 110023 | 12 | 6 | 6 | 2 | 0 |
| cirrhosis | play | 2023 | mult | nll | 1661 | 7905 | 5271 | 18 | 6 | 12 | 3 | 0 |
| concrete strength | play | 2023 | reg | rmse | 765 | 5407 | 3605 | 8 | 0 | 8 | | 0 |
| covertype | play | 2015 | mult | acc | 1692 | 15120 | 565892 | 54 | 44 | 10 | 7 | 0 |
| crab age | play | 2023 | reg | mae | 1429 | 74051 | 49368 | 8 | 1 | 7 | | 0 |
| credit fusion | feat | 2011 | bin | auc | 924 | 150000 | 101503 | 10 | 0 | 10 | 2 | 56384 |
| failure | play | 2022 | bin | auc | 1888 | 26570 | 20775 | 24 | 3 | 21 | 2 | 35982 |
| heat flux fi | play | 2023 | reg | rmse | 693 | 21229 | 10415 | 8 | 2 | 6 | | 34603 |
| horses | play | 2023 | bin | f1 | 1541 | 1235 | 824 | 27 | 17 | 10 | 3 | 1324 |
| housing california | play | 2023 | reg | rmse | 689 | 37137 | 24759 | 8 | 0 | 8 | | 0 |
| influencers | feat | 2013 | bin | auc | 132 | 5500 | 5952 | 22 | 0 | 22 | 2 | 0 |
| insurance | play | 2021 | reg | rmse | 1433 | 300000 | 200000 | 24 | 10 | 14 | | 0 |
| loan approval | play | 2024 | bin | auc | 3858 | 58645 | 39098 | 11 | 4 | 7 | 2 | 0 |
| media | play | 2023 | reg | rmsle | 952 | 360336 | 240224 | 15 | 7 | 8 | | 0 |
| mental health | play | 2024 | bin | acc | 2685 | 140700 | 93800 | 18 | 7 | 8 | 2 | 718167 |
| mercedes | feat | 2017 | reg | r2 | 3823 | 4209 | 4209 | 376 | 376 | 0 | | 0 |
| molecules | feat | 2012 | bin | nll | 698 | 3751 | 2501 | 1776 | 0 | 1776 | 2 | 0 |
| unknown a | play | 2021 | reg | rmse | 1728 | 300000 | 200000 | 14 | 0 | 14 | | 0 |

Table 2: **Metadata of Kaggle challenges.** Challenge types include "playground" (datasets from external sources or synthetically generated) and "featured" (datasets from real scientific or industrial applications, often with significant monetary prizes for top participants). Prediction tasks are binary classification (bin), regression (reg), or multi-class classification (mult; with the number of classes indicated in the #cls column). Note that in our method, *mult* and *bin* are treated the same. Features are categorized as numerical (num) or categorical (cat). The final column reports the number of missing entries in the training data.

## B.2  PER-CHALLENGE RESULTS

| Kaggle Challenge | Meta -Informed | Zero -Shot | Context -Random | Random -Hyperopt | MaxUCB -Hyperopt | LGBM -Hyperopt |
|---|---|---|---|---|---|---|
| abalone | 85.73 ± 3.3 | 74.67 ± 4.6 | 87.87 ± 2.3 | 58.95 ± 4.6 | 56.53 ± 9.0 | 64.21 ± 11.3 |
| allstate | 69.92 ± 2.3 | 61.66 ± 2.9 | 65.41 ± 5.0 | 50.05 ± 2.4 | 56.25 ± 2.7 | 51.0 ± 2.7 |
| attrition | 59.51 ± 1.7 | 61.12 ± 1.8 | 57.31 ± 2.3 | 59.36 ± 3.3 | 58.69 ± 2.6 | 48.21 ± 5.0 |
| blueberry | 81.16 ± 2.4 | 79.86 ± 1.7 | 78.96 ± 3.8 | 62.9 ± 7.1 | 62.9 ± 7.1 | 65.87 ± 7.7 |
| churn | 70.35 ± 0.9 | 68.73 ± 0.9 | 68.71 ± 3.0 | 65.07 ± 4.0 | 62.98 ± 6.0 | 70.64 ± 1.0 |
| cirrhosis | 70.58 ± 3.6 | 69.09 ± 1.4 | 73.06 ± 1.8 | 64.61 ± 4.6 | 66.96 ± 1.9 | 70.17 ± 2.0 |
| concrete strength | 74.34 ± 17.9 | 74.19 ± 6.8 | 59.37 ± 16.1 | 88.81 ± 5.4 | 75.46 ± 13.8 | 83.21 ± 9.3 |
| covertype | 67.78 ± 4.0 | 58.35 ± 7.6 | 60.05 ± 10.3 | 56.75 ± 11.0 | 53.75 ± 6.2 | 32.0 ± 3.4 |
| crab age | 68.87 ± 0.7 | 68.81 ± 0.6 | 67.67 ± 1.2 | 61.84 ± 2.3 | 59.53 ± 3.2 | 63.84 ± 1.8 |
| credit fusion | 96.61 ± 1.0 | 96.71 ± 1.1 | 90.91 ± 1.7 | 96.35 ± 0.9 | 94.12 ± 1.8 | 96.75 ± 1.5 |
| failure | 41.12 ± 1.5 | 43.52 ± 1.7 | 41.25 ± 0.8 | 43.7 ± 2.6 | 47.15 ± 5.0 | 48.15 ± 7.0 |
| heat flux fi | 93.4 ± 5.0 | 90.7 ± 4.3 | 83.65 ± 8.6 | 69.07 ± 6.6 | 47.37 ± 11.3 | 36.22 ± 17.1 |
| horses | 82.39 ± 7.7 | 82.78 ± 5.6 | 75.31 ± 10.6 | 81.15 ± 6.2 | 72.7 ± 9.2 | 79.75 ± 5.7 |
| housing california | 62.53 ± 0.6 | 54.84 ± 2.4 | 60.07 ± 2.0 | 46.9 ± 6.8 | 42.15 ± 8.2 | 52.71 ± 3.9 |
| influencers | 76.84 ± 7.4 | 83.55 ± 1.4 | 80.52 ± 2.8 | 82.95 ± 2.7 | 82.03 ± 3.0 | 87.45 ± 1.9 |
| insurance | 74.68 ± 2.4 | 68.16 ± 1.8 | 67.9 ± 2.1 | 62.53 ± 5.9 | 66.76 ± 4.2 | 64.6 ± 3.4 |
| loan approval | 71.58 ± 2.6 | 63.29 ± 5.5 | 66.84 ± 5.4 | 62.64 ± 6.9 | 60.81 ± 4.8 | 74.43 ± 0.9 |
| media | 62.95 ± 1.4 | 57.52 ± 2.0 | 61.81 ± 2.5 | 49.5 ± 7.5 | 47.87 ± 5.6 | 26.07 ± 2.8 |
| mental health | 92.99 ± 3.0 | 79.77 ± 10.2 | 89.69 ± 5.2 | 75.34 ± 9.5 | 73.39 ± 9.3 | 80.11 ± 7.7 |
| mercedes | 17.81 ± 2.8 | 36.44 ± 7.8 | 35.26 ± 10.6 | 36.57 ± 8.6 | 38.94 ± 4.7 | 25.42 ± 2.0 |
| molecules | 97.52 ± 1.5 | 96.34 ± 1.6 | 96.32 ± 3.3 | 96.33 ± 2.6 | 94.84 ± 1.9 | 78.02 ± 12.6 |
| unknown a | 80.56 ± 0.8 | 78.6 ± 0.8 | 72.59 ± 2.4 | 66.17 ± 2.5 | 61.75 ± 6.0 | 61.41 ± 5.5 |
| **Mean** | **72.69 ± 0.2** | 70.39 ± 0.2 | 70.02 ± 0.3 | 65.7 ± 1.1 | 62.86 ± 1.2 | 61.8 ± 1.1 |

Table 3: Kaggle p-rank results across all challenges (the higher, the better). Uncertainty is reported as ± values, representing the 90% confidence interval based on the standard error across 8 random seeds.

| Kaggle Challenge | Context Blends |
|------------------|----------------|
| abalone | $92.06 \pm 0.1$ |
| allstate | $77.15 \pm 0.7$ |
| attrition | $57.47 \pm 3.2$ |
| blueberry | $88.65 \pm 0.8$ |
| churn | $71.48 \pm 1.1$ |
| cirrhosis | $83.62 \pm 2.7$ |
| concrete strength | $95.95 \pm 2.8$ |
| covertype | $77.16 \pm 1.0$ |
| crab age | $71.51 \pm 0.2$ |
| credit fusion | $97.93 \pm 0.8$ |
| failure | $38.87 \pm 2.9$ |
| heat flux fi | $99.3 \pm 0.1$ |
| horses | $73.73 \pm 12.0$ |
| housing california | $71.57 \pm 1.0$ |
| influencers | $74.24 \pm 1.9$ |
| insurance | $84.46 \pm 6.5$ |
| loan approval | $78.55 \pm 0.9$ |
| media | $72.0 \pm 0.6$ |
| mental health | $75.03 \pm 5.2$ |
| mercedes | $59.43 \pm 4.8$ |
| molecules | $83.63 \pm 12.2$ |
| unknown a | $86.06 \pm 1.4$ |
| **Mean** | $77.72 \pm 0.2$ |

Table 4: Kaggle p-rank results across all challenges (the higher, the better) for **Context Blends**. Uncertainty is reported as $\pm$ values, representing the 90% confidence interval based on the standard error across 8 random seeds.

Looking at the detailed per-challenge results (Tables 3 and 4) alongside the dataset metadata (Table 2), we observe that performance patterns vary across tasks. The **Meta-Informed** method generally performs best on large datasets, particularly in regression tasks, while showing reduced effectiveness on small or extremely "wide" datasets (i.e., those with a high feature-to-sample ratio). On average, it achieves the highest baseline performance with a mean p-rank of 72.69, outperforming **Zero-Shot** (70.39) and standard hyperparameter optimization methods such as **LGBM-Hyperopt** (61.8), though still below the oracle-like **Context Blends** (77.72). Its strongest results are observed in datasets with tens or hundreds of thousands of samples (e.g., `mental health`, `media`, `insurance`, `allstate`) and in regression problems such as `heat flux fi` and `housing california`, where it consistently outperforms other methods by a large margin. Furthermore, it proves robust in handling datasets with missing values, provided they are sufficiently large. In contrast, its performance is more limited on smaller datasets (e.g., `influencers`, `concrete strength`) and it is less competitive on wide datasets with disproportionately many features compared to samples (e.g., `mercedes`, `molecules`). In summary, **Meta-Informed** is particularly well suited for large-scale regression settings with ample training data, while offering more modest gains in low-sample or high-dimensional feature spaces. Notably, while **LGBM-Hyperopt** is the weakest overall baseline, it still achieves top performance on a few datasets (e.g., `influencers`, `concrete strength`), illustrating that in some cases restraining the search space to a single strong predictor can be advantageous.

# C  PROMPTING SCHEMAS

## C.1  CURRENT TASK DESCRIPTION FORMAT

For both prompting strategies, the LLM receives the current task description in the following structured format. Below is an example for the Abalone challenge:

```
# Metadata for kaggle_abalone

## name
kaggle_abalone

## prediction_type
regression

## score_name
rmsle

## n_train: 90615    n_test: 60411    total_samples: 151026    train_test_ratio: 1.5

## features
total: 9    numeric: 8    numerical_range_avg: 11327.82    categorical: 1

### unique_values_per_categorical
min: 3    max: 3    median: 3    mode: 3

## missing_data
has_missing: False    total_missing_values: 0    data_density: 1.0

## target_values
min: 1    max: 29    mean: 9.697    median: 9.0    std: 3.176    skewness: 1.204    kurtosis: 2.613
```

## C.2 ZERO-SHOT SETTING

The following system prompt is used for the Zero-Shot setting.

> **Zero-Shot System Prompt**
>
> You are a data science expert specializing in model blending. You will receive a description of a machine learning tasks and dataset. Your task is to propose a new model blend with exactly 10 models by completing a given JSON file that describes a new task, maintaining the same format. You must output the json with 10 different choices of models and "models" as a key following exactly the input format JSON but removing the prank and mean score columns. Select models and hyperparameters considering factors such as dataset characteristics and task type. Don't forget to give exactly 10 different variations and use the given format for the output adding the needed values lists. A predefined hyperparameter grid will be provided beforehand. Ensure your selections of the 10 models adhere to the available hyperparameter choices and that the number of models given is 10.

In the Zero-Shot setting, the LLM is not provided with in-context examples. To guide its output, it is instead given the expected JSON schema, as shown below.

```
{
  "models": {
    "catboost": {
      "columns": ["bootstrap_type", "border_count", "grow_policy", "l2_leaf_reg", "
          learning_rate",
      "max_depth", "min_data_in_leaf", "n_estimators", "random_strength"],
      "values": []
    },
    "lgbm": {
      "columns": ["boosting_type", "colsample_bynode", "colsample_bytree", "drop_rate",
      "learning_rate", "max_bin", "max_depth", "min_child_weight", "min_data_in_leaf",
      "min_split_gain", "n_estimators", "reg_alpha", "reg_lambda", "subsample"],
      "values": []
    },
    "xgboost": {
      "columns": ["colsample_bylevel", "colsample_bynode", "colsample_bytree", "gamma",
      "learning_rate", "max_depth", "min_child_weight", "n_estimators", "reg_alpha", "
          reg_lambda",
      "subsample"],
      "values": []
    },
    "skmlp": {
      "columns": ["activation", "alpha", "beta_1", "beta_2", "epsilon", "layers",
      "learning_rate_init", "max_iter", "n_iter_no_change", "n_knots"],
      "values": []
    }
  }
}
```

## C.3 META-INFORMED SETTING

The following system prompt is used for the Meta-Informed setting.

---

**Meta-Informed System Prompt**

You are a data science expert specializing in model blending. You will receive multiple descriptions of machine learning tasks, datasets, and the top 10 performing models in an blend in JSON format, including their individual mean scores and percentile ranks. Your task is to propose a new model blend with exactly 10 models by completing a given JSON file that describes a new task, maintaining the same format. You must output the json with 10 different choices of models and "models" as a key following exactly the input format JSON but removing the prank and mean score columns. Select models and hyperparameters based on the similarity between the new task and previous ones, considering factors such as dataset characteristics, task type, and model performance trends. Don't forget to give exactly 10 different variations and use the same format as the input. A predefined hyperparameter grid will be provided beforehand. Ensure your selections of the 10 models adhere to the available hyperparameter choices and that the number of models given is 10.

---

The prompt is enriched with information from other challenges, provided as pairs of task metadata and the top 10 models in the solution blend, formatted as previously described. For each given challenge, we include all other challenges of the same type (classification or regression).

## D CHAT API CONFIGURATION AND DEFAULTS

To invoke the DeepSeek-R1 API, we use the `client.chat.completions.create` function from the OpenAI SDK with temperature $T = 1.0$ and default hyperparameters.

For more information, refer to the official documentation:

- OpenAI Platform: https://platform.openai.com/

- DeepSeek API Docs: https://api-docs.deepseek.com/

## E EXAMPLE REASONING TRACES

To illustrate the model's decision-making process, we show a few selected excerpts from its reasoning traces on different Kaggle tasks.

## Mercedes Dataset (Zero-Shot)

Okay, so I need to create a new model blend for the `Kaggle Mercedes` regression task. The dataset has 377 features, mostly categorical (376) and one numeric. The target is continuous with some skewness. The goal is to propose 10 different models across CatBoost, LGBM, XGBoost, and SKMLP, using the given hyperparameters.

First, let's think about the dataset characteristics. Since there are a lot of categorical features, tree-based models like CatBoost, LGBM, and XGBoost might perform well because they handle categorical variables effectively. `skmlp` might need more tuning but could add diversity to the blend.

For CatBoost, since it's good with categorical data, I'll focus on parameters that handle that. `bootstrap_type` could be Bayesian variants for better handling. `border_count` might be higher (like 512 or 1024) to capture more splits. `grow_policy` could be `Lossguide` for deeper trees. `learning_rate` should be moderate, maybe 0.05 to 0.1. L2 regularization (`l2_leaf_reg`) around 3.0 to prevent overfitting. `n_estimators` around 500–700 for sufficient depth. `random_strength` maybe 1.0 or 5.0 for some randomness.

For LGBM, `boosting_type` like `goss` or `dart` could be efficient. Since there are many features, `colsample_bytree` and `subsample` around 0.8 to reduce overfitting. `learning_rate` similar to CatBoost. `max_depth` around 7–10. `min_data_in_leaf` higher (like 50) given the dataset size. `n_estimators` around 300–500. Regularization parameters (`reg_alpha` and `reg_lambda`) around 0.5–2.0.

XGBoost would need similar considerations. `colsample_bytree` and `subsample` around 0.8. `learning_rate` 0.05–0.1. `max_depth` 6–10. `gamma` around 0.5 to control splits. `reg_alpha` and `reg_lambda` similar to LGBM. `n_estimators` 300–500.

`skmlp` might have layers like `512-256` to handle high dimensionality. `activation=relu` for non-linearity. `learning_rate_init=0.01`. `alpha` (L2 regularization) 0.001 or 0.01. Beta parameters (Adam optimizer) default to 0.9 and 0.999. `max_iter` high (20000) since MLPs need time. `n_iter_no_change=10`. `n_knots` maybe 10 for spline layers.

I need to ensure 10 models in total. Maybe distribute them as 3 CatBoost, 3 LGBM, 3 XGBoost, and 1 SKMLP for diversity. Each model should have unique hyperparameters within the grid. Check each parameter against the allowed values. For example, CatBoost's `bootstrap_type` can't be "No" but has Bayesian options. Ensure all selected values are in the provided grid. Also, no duplicates in the models. Each entry must have distinct hyperparameter combinations.

## Abalone Dataset (Meta-Informed)

Okay, I need to create a new model blend for the `kaggle_abalone` task. Let me start by understanding the problem. The task is regression with RMSLE as the score. The dataset has 9 features, 8 numeric and 1 categorical with 3 unique values. The target has a mean around 9.7, with some skewness and kurtosis. The data is clean with no missing values.

Looking at the past tasks, `kaggle_crab_age` seems similar. Both have regression, a categorical feature with 3 unique values, and similar target distributions (mean around 9–10, skewness > 1). In `kaggle_crab_age`, the top models were CatBoost and LGBM. For example, CatBoost used `bootstrap_type=Bernoulli`, high `border_count=1024`, `grow_policy=Lossguide`, and `n_estimators` around 500–700. LGBM models had `boosting_type=gbdt_1` or `gbdt_5`, high `n_estimators` (1000), and specific regularization parameters.

Also, `kaggle_unknown_a` had a similar setup with all numeric features and high `n_estimators`. XGBoost models there used `colsample_bytree` around 0.5–0.8 and `learning_rate` 0.01–0.1. SKMLP models used `activation=logistic`, low `alpha=0.0001`, and layers such as `256-128`.

Considering hyperparameters from the grid: for CatBoost, possible `bootstrap_type` values include Bernoulli or Bayesian, `border_count=1024`, `grow_policy=Lossguide` or `Depthwise`, `l2_leaf_reg` around 0.2–3.0, `learning_rate=0.1`, `max_depth` around 9–12, and `min_data_in_leaf=700`. For LGBM, suitable choices are `boosting_type=gbdt_1` or `gbdt_5`, `colsample_bynode` 0.8–1.0, `learning_rate=0.05`, `max_depth` around 90, and `min_data_in_leaf=50`. XGBoost could use `colsample_bytree` 0.5–0.8, `gamma` 0.0–1.5, `learning_rate` 0.05–0.1, `max_depth=9`, and `n_estimators=1000`. For SKMLP, I will use `activation=logistic`, `alpha=0.0001`, layers `256-128`, and `learning_rate_init=0.001`.

I need to select 10 models across different algorithms. Let's pick 4 CatBoost, 3 LGBM, 2 XGBoost, and 1 SKMLP, ensuring hyperparameters are drawn from the predefined grid. Check each parameter against the allowed values (e.g., CatBoost `bootstrap_type`, LGBM `boosting_type`), and ensure `n_estimators` is valid. Each model must be unique in its parameter combination. Finally, I will compile the JSON structure without `mean_score` and `prank`, keeping the same format as the input examples.

## Heat Flux Fi Dataset (Meta-Informed)

Okay, I need to create a new model blend for the `kaggle_heat_flux_fi` task. Let's start by understanding the new task's characteristics. The prediction type is regression, and the score is RMSE. The dataset has 21,229 training samples and 10,415 test samples. There are 9 features: 7 numeric and 2 categorical. The numeric features have an average range of about 6,121, which is similar to some previous tasks like `kaggle_abalone` or `kaggle_crab_age`. The categorical features have 4 and 11 unique values, which is a moderate number. There's missing data with a data density of 0.88, which means some imputation might be needed. The target has a mean close to zero, a negative skewness, and high kurtosis, indicating a peaky distribution with a left tail.

Looking at past tasks, models that performed well on datasets with mixed numeric and categorical features, especially those using CatBoost and LGBM, might be relevant. For example, `kaggle_abalone` used CatBoost with `grow_policy=Lossguide` and high `border_count`, which handles categorical features well. Also, tasks with missing data like `kaggle_media` had LGBM models with certain parameters.

The hyperparameter grids need to be considered. For CatBoost, handling missing data is inherent, so models with `bootstrap_type` like Bayesian or MVS might be good. LGBM can handle missing values too. XGBoost doesn't handle categoricals natively, so maybe less emphasis here unless the categoricals are encoded. Since the new task has categorical features, CatBoost and LGBM are probably better choices.

Looking at the previous top models, for regression tasks with similar features, CatBoost models with higher `border_count` (like 1024) and LGBM with higher `max_depth` (like 40, 90) and `n_estimators` (1000) performed well. Also, models with lower learning rates (0.1, 0.15) and regularization (`l2_leaf_reg` around 0.5–2.0) were common.

Considering the new task's missing data, models that handle missing values well are crucial. CatBoost's `bootstrap_type` options like Bayesian or MVS, and LGBM with `drop_rate` and `subsample` parameters could be beneficial. Also, since the target has a negative skew, maybe models that are robust to such distributions, like those with appropriate loss functions or regularization.

Now, selecting 10 models. Let's check the hyperparameter grids to ensure valid choices. For CatBoost, possible `bootstrap_type`: Bayesian_0, MVS, Bernoulli. `border_count` options: 32, 64, 128, etc. For LGBM, `boosting_type`: gbdt_0, dart_1, goss. Also, `subsample` and `colsample` parameters. `skmlp` might be less prioritized due to the categorical features, but maybe include one if previous tasks with similar data used it.

Looking at the previous models, in `kaggle_unknown_a`, CatBoost with `bootstrap_type=No` and `grow_policy=Depthwise` was used. For the new task, maybe mix different bootstrap types. For LGBM, models with higher `n_estimators` (1000) and `max_depth` (like 40, 90) from `kaggle_abalone` and `kaggle_housing_california`.

Also, in the new task, the data has missing values, so models that handle missing data without imputation (like CatBoost and LGBM) are preferred. XGBoost requires handling missing data externally, so maybe less emphasis unless parameters are adjusted, but the hyperparameter grid doesn't include handling for missing data, so maybe avoid XGBoost unless necessary.

Looking at `skmlp`, in past tasks like `kaggle_insurance`, `activation=logistic` with certain layers was used. Maybe include one `skmlp` model for diversity.

Now, compiling 10 models.

# F  BASE MODEL DETAILS

We use four base models in our experiments: XGBOOST (Chen & Guestrin, 2016), CATBOOST (Prokhorenkova et al., 2018), LGBM, and SKMLP (Pedregosa et al., 2011). The corresponding hyperparameter grids for each model are provided in Figure 6.

### CatBoost hyperparameter grid.

```
n_estimators = Hyperparameter(dtype='int', default=400, values=[10, 20, 30, 40, 50, 70, 100, 150, 200, 250, 300, 400, 500, 700, 1000])#, 2000, 3000, 5000, 7000, 10000])
learning_rate = Hyperparameter(dtype='float', default=0.05, values=[0.0005, 0.001, 0.002, 0.005, 0.01, 0.05, 0.1, 0.15, 0.2, 0.25, 0.3, 0.4, 0.5, 0.6, 0.7, 0.8, 0.9, 1.0])
max_depth = Hyperparameter(dtype='int', default=5, values=[1, 2, 3, 4, 5, 6, 7, 8, 9, 10, 11, 12, 13, 14, 15, 16])
l2_leaf_reg = Hyperparameter(dtype='float', default=3.0, values=[0.0, 0.1, 0.2, 0.3, 0.4, 0.5, 1.0, 2.0, 3.0, 4.0, 5.0])
border_count = Hyperparameter(dtype='int', default=254, values=[32, 64, 128, 254, 512, 1024])
grow_policy = Hyperparameter(dtype='str', default='SymmetricTree', values=['SymmetricTree', 'Depthwise', 'Lossguide'])
min_data_in_leaf = Hyperparameter(dtype='int', default=1, values=[1, 5, 10, 20, 50, 100, 200, 500, 700])
bootstrap_type = Hyperparameter(dtype='str', default='No', values=['No', 'Bernoulli', 'MVS', 'Bayesian_0', 'Bayesian_1', 'Bayesian_5', 'Bayesian_10', 'Bayesian_20', 'Bayesian_50'])
random_strength = Hyperparameter(dtype='float', default=1, values=[0, 1, 5, 10, 20, 50, 100])
```

### LGBM hyperparameter grid.

```
colsample_bytree = Hyperparameter(dtype='float', default=0.5, values=[0.1, 0.2, 0.3, 0.4, 0.5, 0.6, 0.7, 0.8, 0.9, 1.0])
colsample_bynode = Hyperparameter(dtype='float', default=0.5, values=[0.1, 0.2, 0.3, 0.4, 0.5, 0.6, 0.7, 0.8, 0.9, 1.0])
min_split_gain = Hyperparameter(dtype='float', default=0.0, values=[0.0, 0.1, 0.2, 0.3, 0.4, 0.5, 0.6, 0.7, 0.8, 0.9, 1.0, 1.2, 1.5, 2.0])
learning_rate = Hyperparameter(dtype='float', default=0.05, values=[0.0005, 0.001, 0.002, 0.005, 0.01, 0.05, 0.1, 0.15, 0.2, 0.25, 0.3, 0.4, 0.5, 0.6, 0.7, 0.8, 0.9, 1.0])
max_depth = Hyperparameter(dtype='int', default=5, values=[1, 2, 3, 4, 5, 6, 7, 8, 9, 10, 12, 14, 16, 18, 20, 25, 30, 35, 40, 50, 60, 70, 80, 90, 100])
min_child_weight = Hyperparameter(dtype='float', default=9, values=[1, 2, 3, 4, 5, 6, 7, 8, 9, 10])
n_estimators = Hyperparameter(dtype='int', default=400, values=[10, 20, 30, 40, 50, 70, 100, 150, 200, 250, 300, 400, 500, 700, 1000])#, 2000, 3000, 5000, 7000, 10000])
reg_alpha = Hyperparameter(dtype='float', default=2.0, values=[0.0, 0.1, 0.2, 0.3, 0.4, 0.5, 1.0, 2.0, 3.0])
reg_lambda = Hyperparameter(dtype='float', default=5.0, values=[0.5, 1.0, 1.5, 2.0, 3.0, 4.0, 5.0])
subsample = Hyperparameter(dtype='float', default=0.9, values=[0.5, 0.6, 0.7, 0.8, 0.9, 1.0])
max_bin = Hyperparameter(dtype='int', default=256, values=[256, 512, 1024, 2048, 4096, 8192])
min_data_in_leaf = Hyperparameter(dtype='int', default=1, values=[1, 5, 10, 20, 50, 100, 200, 500, 700])
boosting_type = Hyperparameter(dtype='str', default='gbdt_5', values=['gbdt_0', 'gbdt_1', 'gbdt_5', 'gbdt_10', 'dart_0', 'dart_1', 'dart_5', 'dart_10', 'goss'])
drop_rate = Hyperparameter(dtype='float', default=0.1, values=[0.1, 0.2, 0.3, 0.4, 0.5, 0.6, 0.7, 0.8, 0.9])
```

### XGBoost hyperparameter grid.

```
colsample_bytree = Hyperparameter(dtype='float', default=0.5, values=[0.1, 0.2, 0.3, 0.4, 0.5, 0.6, 0.7, 0.8, 0.9, 1.0])
colsample_bylevel = Hyperparameter(dtype='float', default=0.5, values=[0.1, 0.2, 0.3, 0.4, 0.5, 0.6, 0.7, 0.8, 0.9, 1.0])
colsample_bynode = Hyperparameter(dtype='float', default=0.5, values=[0.1, 0.2, 0.3, 0.4, 0.5, 0.6, 0.7, 0.8, 0.9, 1.0])
gamma = Hyperparameter(dtype='float', default=0.0, values=[0.0, 0.1, 0.2, 0.3, 0.4, 0.5, 0.6, 0.7, 0.8, 0.9, 1.0, 1.2, 1.5, 2.0])
learning_rate = Hyperparameter(dtype='float', default=0.1, values=[0.0005, 0.001, 0.002, 0.005, 0.01, 0.05, 0.1, 0.15, 0.2, 0.25, 0.3, 0.4, 0.5, 0.6, 0.7, 0.8, 0.9, 1.0])
max_depth = Hyperparameter(dtype='int', default=2, values=[1, 2, 3, 4, 5, 6, 7, 8, 9, 10, 12, 14, 16, 18, 20, 25, 30, 35, 40, 50, 60, 70, 80, 90, 100])
min_child_weight = Hyperparameter(dtype='int', default=1, values=[1, 2, 3, 4, 5, 6, 7, 8, 9, 10])
n_estimators = Hyperparameter(dtype='int', default=700, values=[10, 20, 30, 40, 50, 70, 100, 150, 200, 250, 300, 400, 500, 700, 1000])#, 2000, 3000, 5000, 7000, 10000])
reg_alpha = Hyperparameter(dtype='float', default=0.1, values=[0.0, 0.1, 0.2, 0.3, 0.4, 0.5, 1.0, 2.0, 3.0])
reg_lambda = Hyperparameter(dtype='float', default=0.5, values=[0.5, 1.0, 1.5, 2.0, 3.0, 4.0, 5.0])
subsample = Hyperparameter(dtype='float', default=1.0, values=[0.5, 0.6, 0.7, 0.8, 0.9, 1.0])
```

### SKMLP hyperparameter grid.

```
layers = Hyperparameter(dtype='str', default='512-256', values=['32', '64', '128', '256', '256-128', '512-256'])#, '1024-512', '1024-512-512'])
activation = Hyperparameter(dtype='str', default='tanh', values=['relu', 'tanh', 'logistic'])
alpha = Hyperparameter(dtype='float', default=0.1, values=[0.0001, 0.001, 0.01, 0.1])
learning_rate_init = Hyperparameter(dtype='float', default=0.01, values=[0.001, 0.01, 0.1])
max_iter = Hyperparameter(dtype='int', default=5000, values=[5000, 10000, 20000])
n_iter_no_change = Hyperparameter(dtype='int', default=10, values=[5, 10, 20])
beta_1 = Hyperparameter(dtype='float', default=0.8, values=[0.8, 0.9, 0.95])
beta_2 = Hyperparameter(dtype='float', default=0.999, values=[0.99, 0.999, 0.9999])
epsilon = Hyperparameter(dtype='float', default=1e-7, values=[1e-8, 1e-7, 1e-6])
n_knots = Hyperparameter(dtype='int', default=5, values=[3, 5, 10, 20])
```

Figure 6: Base models hyperparameters.

# G  BASELINES DESCRIPTION

## G.1  CONTEXT-RANDOM

For the **Context-Random** baseline, we uniformly sample $n$ model–hyperparameter configurations from the same pool of prior-task blends that are provided as context in the **Meta-Informed** setting. This isolates whether improvements come from meaningful adaptation by the LLM or simply from re-using high-quality configurations already present in the context.

We fix $n = 10$ to match the number of configurations proposed by the LLM in a single run.

## G.2  RANDOM-HYPEROPT

For the **Random-Hyperopt** baseline, we use HEBO to optimize hyperparameters within a model family, but the model family itself is selected uniformly at random at each round. Concretely, at each iteration one of the base learners is sampled with equal probability, after which HEBO proposes a new configuration for that family. This ensures a simple exploration strategy without bias toward any particular model type.

## G.3  LGBM-HYPEROPT

For the **LGBM-Hyperopt** baseline, we restrict the search space to the LightGBM model family. At each evaluation round, we apply the HEBO optimizer to propose a new LightGBM configuration, which is then trained and evaluated on the target dataset. This baseline isolates the performance

of hyperparameter optimization when applied to a single strong gradient boosting method without model family selection. As with the other baselines, we allocate a fixed budget of 10 evaluations when comparing against the LLM recommendations.

### G.4    MAXUCB-HYPEROPT

For the **MaxUCB-Hyperopt** baseline, we implement the bandit-based CASH formulation proposed by Balef et al. (2025). In this setting, each candidate model family is treated as an arm in a multi-armed bandit, and hyperparameter optimization is carried out within the selected arm using HEBO. The Max-UCB algorithm balances exploration of new model families with exploitation of those that have already demonstrated promising performance.

At each round $t$, the utility of arm $i$ is computed as:

$$U_i = \max(r_{i,1}, \ldots, r_{i,n_i}) + \left( \frac{\alpha \log(t)}{n_i} \right)^2,$$

where $r_{i,j}$ denotes the observed rewards (validation scores) from the $j$-th configuration of model family $i$, and $n_i$ is the number of configurations tried so far for that family. The algorithm selects the arm

$$I_t = \arg \max_{i \leq K} U_i,$$

applies HEBO within that model family to propose a new hyperparameter configuration, and observes the resulting reward.

Following recommendations from the original paper, we set the exploration parameter to $\alpha = 0.5$, which provides a favorable balance between exploration and exploitation across tasks.

## H ROBUSTNESS TO PROMPT SHUFFLING

Large language models can sometimes exhibit position or recency biases (Wang et al., 2023; 2025), raising the question of whether the **Meta-Informed** strategy is sensitive to the way information is ordered inside the prompt. To test this, we generate two independent shuffled versions of the Meta-Informed prompt for each dataset–seed pair. In each shuffle, we randomly permute (i) the order of support datasets, (ii) the order of model families listed in the schema, and (iii) the order of hyperparameters within each family. The underlying content is unchanged, only the presentation order differs. The experimental setup is otherwise identical on the 22 Kaggle datasets the same contexts, ensembling pipeline, and $p_{\text{rank}}$ as the evaluation metric.

**Results.** Across 22 paired comparisons, we observe no statistically significant difference between the two shuffled versions (paired t-test: $t = -1.48$, $p = 0.153$, $df = 21$). The mean difference in $p_{\text{rank}}$ is $-1.86$ points, indicating that the second shuffle tends to achieve slightly better ranks, though this difference is not significant. The effect size is small (Cohen's $d = -0.32$), and a non-parametric Wilcoxon signed-rank test confirms these findings ($p = 0.149$). Individual challenge results show mixed outcomes, with some favoring each version, consistent with random variation rather than systematic bias.

These results are consistent with the **Meta-Informed** strategy being robust to prompt ordering, with no evidence that the arrangement of elements within the prompt systematically affects performance.

Table 5: Private leaderboard p-rank for two shuffled prompt versions across 22 Kaggle datasets.

| Kaggle Challenge | Shuffle 1 | Shuffle 2 | $\Delta$ (1–2) |
|---|---|---|---|
| abalone | 89.64 | 88.30 | +1.34 |
| allstate | 59.34 | 70.34 | -11.00 |
| attrition | 60.45 | 65.41 | -4.96 |
| blueberry | 89.33 | 88.43 | +0.91 |
| churn | 70.79 | 72.08 | -1.29 |
| cirrhosis | 70.62 | 69.30 | +1.32 |
| concrete strength | 84.58 | 95.82 | -11.24 |
| covertype | 37.65 | 45.21 | -7.57 |
| crab age | 70.26 | 70.26 | 0.00 |
| credit fusion | 95.67 | 96.86 | -1.19 |
| failure | 48.99 | 39.19 | +9.80 |
| heat flux fi | 96.83 | 96.39 | +0.43 |
| housing california | 56.17 | 57.04 | -0.87 |
| horses | 72.23 | 85.85 | -13.63 |
| influencers | 84.85 | 85.61 | -0.76 |
| insurance | 79.83 | 69.85 | +9.98 |
| loan approval | 76.33 | 74.78 | +1.56 |
| media | 59.56 | 67.12 | -7.56 |
| mental health | 96.50 | 98.44 | -1.94 |
| mercedes | 20.35 | 23.10 | -2.75 |
| molecules | 99.71 | 98.28 | +1.43 |
| unknown a | 72.05 | 74.88 | -2.84 |
| **Mean** | 73.43 | 75.29 | -1.86 |

## LLM USAGE STATEMENT

Large language models were used exclusively as assistive tools for minor writing support, such as polishing grammar, improving clarity, and suggesting alternative phrasings. They were not involved in research ideation, experimental design, implementation and analysis. All scientific contributions and conclusions are solely the work of the authors.

