# OpenReview forum: "LLMs as In-Context Meta-Learners for Model and Hyperparameter Selection"
_ICLR.cc/2026/Conference — Submitted to ICLR 2026_

### Official Review · Reviewer_i4zb · 2025-10-22

**Soundness:** 3
**Presentation:** 3
**Contribution:** 1
**Rating:** 2
**Confidence:** 4

**Summary:**

The paper proposes the application of LLMs to the Combined Algorithm Selection and Hyperparameter (CASH) problem. The authors present two prompting strategies: a "Zero-Shot" approach, which relies on a description of the dataset, and a "Meta-Informed" approach, which includes examples of previously solved tasks in the prompt. The performance of these methods is evaluated on a synthetic task and a benchmark of 22 real-world Kaggle datasets, with the goal of demonstrating that LLMs can act as efficient, non-iterative assistants for AutoML.

**Strengths:**

1. Well-presented manuscript: The paper is clearly written, well-structured, and easy to follow. The authors do an excellent job of explaining their methodology and presenting their results in an organized fashion, which aids in understanding the work.

2. Comprehensive benchmark collection: A great effort was clearly invested in collecting and evaluating the method on a diverse set of 22 Kaggle challenges. This provides a broad and transparent, albeit unflattering, view of the proposed method's performance across different real-world tabular data problems.

**Weaknesses:**

1. Lack of novelty.
At its core, this paper does not introduce a new method. It is a straightforward application of standard zero-shot and few-shot prompting to an existing problem. Describing this as "in-context meta-learning" is a generous reframing of what is. The paper offers no new algorithms, no architectural insights, and no novel techniques. Besides, LLM has been applied for AutoML in a more comprehensive way in literature. It is an application study that fails to contribute meaningfully to the advancement of either AutoML or the science of LLMs.

2. The problem scenario is ill-suited for an LLM and the performance is not good enough.
The paper fails to justify why the CASH problem, particularly for a single dataset, is a task that requires an LLM. An experienced data scientist can typically select a strong model and a reasonable hyperparameter search space in minutes by applying common heuristics. The LLM's reasoning traces confirm that it is merely replicating these same simple heuristics (e.g., "CatBoost is good for categorical features"). More importantly, the results are underwhelming. No significant performance gap can be witnessed between applying LLM and some simple baselines like Context-Random. Moreover, no baseline with expert's knowledge nor AutoML in the same search space are compared, which are expected to be likely show better performance than the proposed method.

3. The "Meta-Informed" approach is impractical.
This is the most critical flaw of the paper. The "Meta-Informed" strategy, which drives the best results, requires providing the LLM with "high-performing configurations" (theta*) from past tasks. The authors state these ground-truth solutions were obtained by "running extensive hyperparameter search."
In real-world scenarios where facing a new ML problem,  the access of high-quality examples are not practical.
To use this method, a user must have already invested the time and resources to collecting highly-related and generalizable other tasks/datasets and and get theta* for each of them using exhausting search.
It is a solution that requires you to have already solved the problem many times over, while such scenario may not need the proposed method.

**Questions:**

Please refer to Questions

---

> ### Author Response · Authors · 2025-11-20
> **Rebuttal by Authors**
>
> We thank Reviewer i4zb for their thoughtful comments and evaluation.
>
> >Lack of novelty. At its core, this paper does not introduce a new method. It is a straightforward application of standard zero-shot and few-shot prompting to an existing problem. Describing this as "in-context meta-learning" is a generous reframing of what is. The paper offers no new algorithms, no architectural insights, and no novel techniques. Besides, LLM has been applied for AutoML in a more comprehensive way in literature. It is an application study that fails to contribute meaningfully to the advancement of either AutoML or the science of LLMs.
>
> Our goal is to examine whether pretrained LLMs can act as in-context meta-learners that generate competitive model–hyperparameter configurations under low compute budgets. While the prompting mechanisms themselves are standard, the empirical result that LLMs benefit substantially from Meta-Informed prompting appears to be new in the CASH setting.
>
> >The problem scenario is ill-suited for an LLM and the performance is not good enough. The paper fails to justify why the CASH problem, particularly for a single dataset, is a task that requires an LLM. An experienced data scientist can typically select a strong model and a reasonable hyperparameter search space in minutes by applying common heuristics. The LLM's reasoning traces confirm that it is merely replicating these same simple heuristics (e.g., "CatBoost is good for categorical features"). More importantly, the results are underwhelming. No significant performance gap can be witnessed between applying LLM and some simple baselines like Context-Random. Moreover, no baseline with expert's knowledge nor AutoML in the same search space are compared, which are expected to be likely show better performance than the proposed method.
>
> Our aim is not to claim that CASH for a single dataset requires an LLM or that LLMs surpass experienced practitioners. Rather, we want to show that LLMs can act as in-context meta-learners capable of efficiently generating competitive configurations. The reasoning traces naturally reflect common heuristics, but the fact that Meta-Informed consistently outperforms Zero-Shot and Context-Random baselines indicates that the model is using task metadata rather than applying fixed rules.
> Our focus is on efficiency: with budgets of only 10 model trainings, LLM-generated configurations match or exceed HPO baselines. AutoML systems generally rely on much larger budgets and richer pipelines, making direct comparisons difficult.
> Importantly, we view this approach as complementary: LLM-generated configurations can serve as strong warm-starts for standard AutoML or HPO workflows, potentially reducing their search time while preserving their eventual performance.
>
> >The "Meta-Informed" approach is impractical. This is the most critical flaw of the paper. The "Meta-Informed" strategy, which drives the best results, requires providing the LLM with "high-performing configurations" (theta*) from past tasks. The authors state these ground-truth solutions were obtained by "running extensive hyperparameter search." In real-world scenarios where facing a new ML problem, the access of high-quality examples are not practical. To use this method, a user must have already invested the time and resources to collecting highly-related and generalizable other tasks/datasets and and get theta* for each of them using exhausting search. It is a solution that requires you to have already solved the problem many times over, while such scenario may not need the proposed method.
>
> While the Meta-Informed setting does assume access to configurations from previously handled tasks, these do not need to be obtained through exhaustive search. In practice, prior tasks are often already "solved" or have good-enough configurations available from routine model development, historical experiments, or even lightweight HPO runs. Our goal is to show that, when such prior experience exists as is common in many applied ML settings the LLM can leverage this knowledge to adapt more effectively to a new task.

---

### Official Review · Reviewer_5AAH · 2025-10-27

**Soundness:** 2
**Presentation:** 3
**Contribution:** 2
**Rating:** 4
**Confidence:** 3

**Summary:**

This paper investigates the use of Large Language Models (LLMs) as in-context meta-learners for the Combined Algorithm Selection and Hyperparameter optimization (CASH) problem. The authors propose converting tabular datasets into interpretable textual metadata, which is then fed into an LLM to recommend a complete ensemble of models and their hyperparameters in a single pass. They evaluate two strategies: a "zero-shot" mode relying only on the LLM's pretrained knowledge, and a "meta-informed" mode augmented with in-context examples of high-performing configurations from past tasks. A synthetic ridge experiment demonstrates scale-dependent emergence of in-context meta-learning (with Qwen2.5 variants), and real-world tests on 22 Kaggle tabular challenges show Meta-Informed outperforming several HEBO-based HPO baselines at comparable training budgets using private leaderboard percentile rank.

**Strengths:**

1. The paper introduces a compelling "search-free" paradigm for tackling the CASH problem. Unlike traditional AutoML methods that rely on expensive, iterative search (e.g., Bayesian optimization, evolutionary algorithms), this approach generates a strong set of candidate models in a single forward pass. This has significant practical implications for efficiency and accessibility.
2. The synthetic Ridge regression experiment (Section 4) is a key strength. By using Random Matrix Theory to derive a closed-form, analytic test error, the authors cleverly isolate and validate the LLM's in-context meta-learning capability without the confounding variable of cross-validation noise
3. The evaluation on 22 real-world Kaggle datasets is robust. The choice of private leaderboard percentile rank ($p_{rank}$) as a metric is excellent.

**Weaknesses:**

1. Assumption of access to high-quality “Context Blends” from prior tasks to populate Meta-Informed prompts may be unrealistic in cold-start or non-tabular settings, and may partly bake strong expert/HEBO search into the context rather than purely demonstrating generalization.
2. (my key concern) The paper claims to address the CASH problem, but it has limited model family coverage (CatBoost, LightGBM, XGBoost, scikit-learn MLP), constraining claims of generality across CASH, omitting linear models, tree ensembles beyond GBMs, kNN, and modern deep tabular architectures.
3. There is a growing literature on recommending models in an unsupervised way, which is completely ignored. For example,
[1] Guha, Neel, et al. "Smoothie: Label free language model routing." Advances in Neural Information Processing Systems 37 (2024): 127645-127672.
[2] Jeong, Daniel P., Zachary C. Lipton, and Pradeep Ravikumar. "Llm-select: Feature selection with large language models." arXiv preprint arXiv:2407.02694 (2024).
[3] https://openreview.net/pdf?id=gWi4ZcPQRl
[4] Jiang, Chumeng, et al. "Beyond Utility: Evaluating LLM as Recommender." Proceedings of the ACM on Web Conference 2025. 2025.

**Questions:**

1. In Fig. 1, it is not clear why the graph for log mean goes down with increasing k. This behavior means that the true configuration is actually closer to the mean of the configurations of other tasks. Is there any relation between the k tasks (in context) and the task in hand? From the problem setting, I would not have expected any smooth pattern for the log mean curve.
2. In Fig. 1, except for the Qwen2.5 7B model, all other curves look almost monotonically declining. Any reason for the distinct behavior of a smaller model?
3. How sensitive are results to the exact metadata schema, and which fields matter most; can an ablation quantify the marginal utility of each metadata component and the effect of richer metafeatures (e.g., simple statistics per feature group, target skew/kurtosis, leakage checks)?
4. How well does the approach extend beyond tabular tasks and the four model families used here, and what breaks when adding neural tabular architectures or broader pipelines (e.g., preprocessing, feature engineering, and metric-specific loss surrogates)?

---

> ### Author Response · Authors · 2025-11-20
> **Rebuttal by Authors**
>
> We thank Reviewer 5AAH for their review and the time dedicated to evaluating our work.
>
> > Assumption of access to high-quality “Context Blends” from prior tasks to populate Meta-Informed prompts may be unrealistic in cold-start or non-tabular settings, and may partly bake strong expert/HEBO search into the context rather than purely demonstrating generalization.
>
> Our approach assumes access only to task-level metadata and configurations obtained from prior training tasks, which is a standard requirement for any meta-learning setting. In our experiments, these "Context Blends" are constructed from the data of previous tasks and do not rely on privileged information. We view them as demonstrations rather than expert priors. Moreover, the Meta-Informed prompts outperform both Zero-Shot and Context-Random baselines, indicating that the LLM is using the alignment between metadata and configurations rather than simply inheriting HEBO or expert knowledge.
> To further test this interpretation, we are currently running a new experiment in which metadata–configuration pairings are randomly reassigned. This control isolates the effect of cross-task structure. If performance returns to the Zero-Shot level, it would confirm that the Meta-Informed gains stem from relational reasoning rather than the mere presence of high-quality configurations. This additional analysis will help clarify the generalization behavior of the method.
>
>
> > In Fig. 1, it is not clear why the graph for log mean goes down with increasing k. This behavior means that the true configuration is actually closer to the mean of the configurations of other tasks. Is there any relation between the k tasks (in context) and the task in hand? From the problem setting, I would not have expected any smooth pattern for the log mean curve
>
> The smoothness of the log-mean curve comes from averaging results over 1,000 random seeds. As k grows, the context set becomes a larger and more representative sample of the overall task distribution, so the log-mean predictor converges toward the true global mean of the configuration space. Its performance therefore stabilizes once k is large enough. This behavior reflects a statistical averaging effect rather than any task-specific similarity.
>
>
> > In Fig. 1, except for the Qwen2.5 7B model, all other curves look almost monotonically declining. Any reason for the distinct behavior of a smaller model?
>
> The distinct behavior of Qwen2.5-7B is likely due to its smaller capacity: unlike the larger models, it may not be able to effectively use a large number of in-context examples. When many examples are provided, the model may struggle to extract the relevant task–configuration relationships
>
> > How sensitive are results to the exact metadata schema, and which fields matter most; can an ablation quantify the marginal utility of each metadata component and the effect of richer metafeatures (e.g., simple statistics per feature group, target skew/kurtosis, leakage checks)?
>
> We intentionally used a simple metadata schema to test whether LLMs can perform CASH without relying on heavy feature engineering. Our results show that even this minimal metadata is sufficient for strong task-conditioned adaptation. We agree that analyzing which components matter most and exploring richer metafeatures is a valuable direction.
>
> > How well does the approach extend beyond tabular tasks and the four model families used here, and what breaks when adding neural tabular architectures or broader pipelines (e.g., preprocessing, feature engineering, and metric-specific loss surrogates)?
>
> Our aim in this paper is to show that LLMs can act as in-context meta-learners for the CASH problem. We therefore focused on tabular tasks and a limited set of model families to clearly isolate LLM capabilities. While extending model coverage to neural tabular architectures or broader pipelines is technically straightforward within our framework, its effect on performance would need to be studied.

---

### Official Review · Reviewer_LM2z · 2025-10-31

**Soundness:** 2
**Presentation:** 3
**Contribution:** 2
**Rating:** 2
**Confidence:** 4

**Summary:**

This paper investigates an innovative AutoML paradigm by employing Large Language Models (LLMs) as "In-Context Meta-Learners" to address the computationally expensive and expert-reliant Combined Algorithm Selection and Hyperparameter optimization (CASH) problem. The authors propose that by converting datasets into metadata and prompting an LLM, the model can recommend high-quality configurations in a "one-shot" manner, bypassing traditional iterative search. The study (utilizing both Zero-Shot and Meta-Informed strategies) demonstrates that LLMs, particularly large-scale ones, can not only leverage pre-trained knowledge but also perform effective meta-learning from in-context "support examples," achieving performance superior to traditional HPO baselines under a low-budget setting.

**Strengths:**

1. **Novelty of Approach:** The paper introduces a new paradigm for the CASH problem, treating the LLM as an in-context meta-learner. This "one-shot" recommendation (rather than iterative search) is an important and insightful exploration in contrast to traditional HPO and AutoML methods (e.g., Bayesian Optimization).

2. **Significant Efficiency and Performance:** Under a fixed low-budget (10 model configurations), the Meta-Informed strategy's average performance across 22 Kaggle tasks significantly surpasses several strong baselines (including Random-Hyperopt and MaxUCB-Hyperopt). This demonstrates the method's strong practical value and efficiency advantage in "cold-start" scenarios with limited computational resources.

3. **Strong Evidence for In-Context Meta-Learning:** The paper clearly demonstrates the meta-learning capabilities of LLMs through two experimental dimensions. First, the synthetic experiment (Fig. 2) proves this capability is scale-dependent. Second, on real-world tasks, the Meta-Informed strategy's performance (Fig. 3) significantly exceeds the Context-Random baseline, providing compelling evidence that the LLM is performing effective reasoning and adaptation rather than merely replicating the context.

**Weaknesses:**

**1. Generality of Claims:** The paper's main conclusions are drawn from evaluations on Kaggle tabular tasks and a limited set of four (primarily tree-based) model families. This represents a significant simplification of the full CASH problem. The authors should discuss the challenges of extending this method to broader task domains (e.g., time series, NLP/CV) and more diverse model libraries (e.g., linear models, deep tabular models), as the generality of the current claims is questionable.

**2. Fairness of Evaluation:** The comparison of a one-shot generation of 10 configurations against the first 10 rounds of traditional HPO may be unfair. HPO methods (like HEBO) often require a cold-start phase, and a 10-round budget is likely insufficient to reflect their true performance potential.

**3. Attribution of Gains:** The source of the observed performance gains is unclear. Is the benefit derived from the "quality" of the LLM's recommendations, or merely from the "quantity" of ensembling 10 candidates? The authors should provide ablation studies (e.g., comparing 1, 5, or 20 candidates). Furthermore, the method relies heavily on high-quality metadata, but its robustness in the presence of noisy or incomplete metadata—a common real-world scenario—is not tested.

**4. Lack of Iterative LLM Baselines:** The comparison is currently limited to traditional HPO (HEBO/MaxUCB) and Context-Random. The authors may argue that their "one-shot CASH" setting is distinct from iterative HPO, but the paper lacks a direct comparison against representative *iterative LLM4HPO* methods (e.g., LLAMBO [1], AgentHPO [2], AutoML-Agent [3]). We believe that **while the task settings (CASH vs. HPO) differ slightly, these methods represent the SOTA for LLMs in optimization. A comparison is essential to highlight the true value and boundaries of this work's one-shot meta-learning approach.**

**5. Potential Pre-training Contamination:** A critical concern is overlooked: the public Kaggle tasks used for evaluation are very likely to have been included in the LLMs' pre-training corpora. If so, the model's strong performance (especially in the Zero-Shot mode) may not stem from novel "meta-learning reasoning" but rather from "memory retrieval" of known solutions. The authors must address or conduct experiments (e.g., using private datasets) to rule out this possibility.

**References:**

> Liu, Tennison, et al. "Large Language Models to Enhance Bayesian Optimization." ICLR 2024
>
> Liu, Siyi, et al. "AgentHPO: Large language model agent for hyper-parameter optimization." CPAL 2025
>
> Trirat, Patara, et al. "AutoML-Agent: A Multi-Agent LLM Framework for Full-Pipeline AutoML." ICML 2025

**Questions:**

None

---

> ### Author Response · Authors · 2025-11-20
> **Rebuttal by Authors**
>
> We thank Reviewer LM2z for their helpful feedback and comments.
>
> > 1. Generality of Claims: The paper's main conclusions are drawn from evaluations on Kaggle tabular tasks and a limited set of four (primarily tree-based) model families. This represents a significant simplification of the full CASH problem. The authors should discuss the challenges of extending this method to broader task domains (e.g., time series, NLP/CV) and more diverse model libraries (e.g., linear models, deep tabular models), as the generality of the current claims is questionable.
>
> Our aim in this paper is to show that LLMs can act as in-context meta-learners for the CASH problem. We therefore focused on tabular tasks and a limited model set to clearly isolate LLM capabilities. We agree that extending to other domains and richer model spaces might require more sophisticated metadata, and we can add a brief discussion clarifying these limitations and outlining directions for broader applicability.
>
> > 2. Fairness of Evaluation: The comparison of a one-shot generation of 10 configurations against the first 10 rounds of traditional HPO may be unfair. HPO methods (like HEBO) often require a cold-start phase, and a 10-round budget is likely insufficient to reflect their true performance potential.
>
> Our comparison matches training budgets: the LLM proposes 10 configurations and we train exactly those 10 models, so comparing to the first 10 HPO rounds is fair in terms of actual compute used. We also include an extended efficiency study (Fig. 4) showing that even with 50 iterations, HPO methods typically remain comparable to or below the LLM-based approaches. More broadly, the goal is to demonstrate the usefulness of LLMs as efficient configuration generators that can also serve as strong warm starts for downstream HPO.
>
> > 3. Attribution of Gains: The source of the observed performance gains is unclear. Is the benefit derived from the "quality" of the LLM's recommendations, or merely from the "quantity" of ensembling 10 candidates? The authors should provide ablation studies (e.g., comparing 1, 5, or 20 candidates). Furthermore, the method relies heavily on high-quality metadata, but its robustness in the presence of noisy or incomplete metadata—a common real-world scenario—is not tested.
>
> In all our comparisons, all methods form an ensemble from the full set of models they train, so the gains cannot be explained by simply using more ensemble members. The difference comes from the quality and complementarity of the LLM-proposed configurations.
> Regarding metadata, our approach relies on statistics computed from the training data the test set so the method does not assume privileged information. We agree that assessing robustness to noisier or less informative metadata is an important direction for future work and we can note this limitation in the discussion.

---

> > ### Author Response · Authors · 2025-11-20
> > **Rebuttal by Authors**
> >
> > > Lack of Iterative LLM Baselines: The comparison is currently limited to traditional HPO (HEBO/MaxUCB) and Context-Random. The authors may argue that their "one-shot CASH" setting is distinct from iterative HPO, but the paper lacks a direct comparison against representative iterative LLM4HPO methods (e.g., LLAMBO [1], AgentHPO [2], AutoML-Agent [3]). We believe that while the task settings (CASH vs. HPO) differ slightly, these methods represent the SOTA for LLMs in optimization. A comparison is essential to highlight the true value and boundaries of this work's one-shot meta-learning approach.
> >
> > We agree that comparisons with iterative LLM-based HPO methods could be informative. Our work targets the one-shot CASH setting, where all configurations are produced in a single inference, and we already benchmark against established non-LLM iterative optimizers (HEBO). Iterative LLM frameworks operate in a feedback-driven regime and optimize a different problem formulation; however, comparing their relative efficiency would indeed be an interesting direction.
> >
> >
> > > Potential Pre-training Contamination: A critical concern is overlooked: the public Kaggle tasks used for evaluation are very likely to have been included in the LLMs' pre-training corpora. If so, the model's strong performance (especially in the Zero-Shot mode) may not stem from novel "meta-learning reasoning" but rather from "memory retrieval" of known solutions. The authors must address or conduct experiments (e.g., using private datasets) to rule out this possibility.
> >
> > We acknowledge that some Kaggle datasets may appear in LLM pre-training corpora. However, this possibility does not affect the central empirical findings of the paper. Our primary result concerns the relative improvement of the Meta-Informed setting compared with the Zero-Shot setting. Any pre-training exposure would influence both conditions symmetrically, as they are evaluated on the same tasks with identical metadata. Thus, potential contamination cannot account for the observed advantage of the Meta-Informed approach, which arises from the model’s ability to exploit cross-task structure provided in-context rather than from prior memorization.

---

### Official Review · Reviewer_WDu3 · 2025-10-31

**Soundness:** 3
**Presentation:** 3
**Contribution:** 2
**Rating:** 4
**Confidence:** 4

**Summary:**

The paper proposes using LLMs as in-context meta-learners to address the CASH problem by prompting an LLM with human-interpretable metadata for each dataset to directly recommend model families and full hyperparameter configurations. It studies two prompting regimes: a Zero-Shot mode that relies only on target-task metadata, and a Meta-Informed mode that augments the prompt with pairs of prior tasks' metadata and their well-performing configurations, thereby enabling cross-task generalization without iterative validation feedback. The system instructs the LLM to output a structured (JSON) ensemble of candidate models drawn from a predefined, family-specific grid (CatBoost, LightGBM, XGBoost, MLP), with lightweight post-processing to project any out-of-grid values, and uses these recommendations in a downstream training and blending pipeline. Overall, the results indicate that LLMs can leverage task metadata to make competitive, search-free recommendations, with the Meta-Informed strategy improving over Zero-Shot, suggesting a practical role for LLMs as efficient assistants within AutoML workflows.

**Strengths:**

1. The proposed method is simple and easy to understand, and as shown in the Experiments section, it demonstrates strong performance compared to other baselines for the CASH problem.

2. The synthetic ridge regression experiment appears reasonable and serves as an effective validation of the motivation behind the proposed methodology.

3. The authors evaluated the proposed method on a sufficiently large dataset -- 22 Kaggle tabular challenges spanning both regression and classification -- which provides a solid empirical basis for the study.

**Weaknesses:**

1. The paper appears to lack novelty. The proposed *Meta-informed* approach seems to be a straightforward application of an in-context learning method to the CASH problem. It would be helpful to more clearly articulate what distinguishes this approach from existing *zero-shot* or *in-context learning (ICL)* paradigms.

2. The paper does not sufficiently explain the rationale, intuition, or assumptions underlying why the proposed method works. For example, examining the model’s internal mechanisms from an interpretability perspective could offer valuable insights, as could an analysis from a data perspective. The absence of such deeper analyses weakens the overall contribution.

3. In addition, the paper does not address well-known phenomena in ICL research, such as *example selection* or *order sensitivity*. It would strengthen the work to examine whether these phenomena also arise in the context considered by the paper and to discuss how they can be effectively handled in this task setting.

4. The methodology involves multiple components -- e.g., elements in the task metadata, ensemble models, and the defined search space -- but lacks an ablation study to evaluate the importance of each. Such an analysis would help clarify which components are most critical to the method’s effectiveness.

**Questions:**

1. Have you tried fine-tuning the LLMs using the same data split? It would be great if you could provide the corresponding experimental results.

2. It is well known that ICL is sensitive to factors such as the type and order of in-context examples. Conducting additional experiments to analyze this sensitivity would strengthen the work.

3. Could you elaborate further on the logistic regression described in line 192?

4. In Figure 2, the 32B model appears to underperform compared to the 7B and 14B models. Could you explain why this occurs? To maintain consistency with the claim in Section 4, larger models within the same family should exhibit better performance, but this trend is not observed.

5. How did you define the search space used as input to the LLMs?

6. I would suggest adopting a constrained decoding strategy to prevent malformed output generation.

7. If there is a recent state-of-the-art (SOTA) method for the CASH problem, please mention it and compare your approach against it.

---

> ### Author Response · Authors · 2025-11-20
> **Rebuttal by Authors**
>
> We appreciate Reviewer WDu3's comments and the time spent evaluating our work.
>
> > Have you tried fine-tuning the LLMs using the same data split? It would be great if you could provide the corresponding experimental results.
>
> Thank you for the suggestion. We have not performed fine-tuning experiments. While this is an interesting direction, our current dataset contains only 22 training tasks, which might be too small to fine-tune an LLM.
>
> > It is well known that ICL is sensitive to factors such as the type and order of in-context examples. Conducting additional experiments to analyze this sensitivity would strengthen the work.
>
> On this point, we’d like to note that Appendix H already includes an analysis of example order. For each dataset–seed pair, we generated two independently shuffled versions of the Meta-Informed prompt by randomizing the order of support datasets, model families, and hyperparameters, without changing any content. Evaluated on the Kaggle datasets, the two shuffled prompts showed no statistically significant difference in $p_\text{rank}$ under either a paired t-test or Wilcoxon test. Overall, we didn’t find evidence that prompt ordering systematically affects the method’s performance.
>
> > Could you elaborate further on the logistic regression described in line 192?
>
> We used the scikit-learn logistic regression classifier. Its inputs are vectors encoding the task characteristics (as defined in the paper), $\mathcal{T} = (n_1, n_2, \mu_1, \mu_2, \alpha_1, \alpha_2)$, and it outputs a prediction $\hat{\lambda}$ selected from the same grid used for the other baselines, $\Lambda = {10^{-4}, 10^{-3}, \ldots, 10^{3}}$.
> The model is trained on $k$ "solved" context tasks, each labeled with its corresponding $\lambda^*$, the value from the grid that minimizes the analytic test error.
>
>
> > In Figure 2, the 32B model appears to underperform compared to the 7B and 14B models. Could you explain why this occurs? To maintain consistency with the claim in Section 4, larger models within the same family should exhibit better performance, but this trend is not observed.
>
> In Section 4, we do not claim that larger models within a family necessarily achieve better overall performance. Our point is that the ability to effectively use task context only emerges once the model is above a certain scale. In practice, however, this does not guarantee monotonic improvements across 7B, 14B, and 32B variants. As shown in Figure 2, all three models still perform close to simple baselines.
>
> > How did you define the search space used as input to the LLMs?
>
> The search space corresponds to a grid of hyperparameters values that we manually defined based on the meaning and usual range of each hyperparameter.
>
> > I would suggest adopting a constrained decoding strategy to prevent malformed output generation.
>
> Given that the rate of malformed outputs was low, we did not feel the need to constrain decoding. However, we agree that such constraints could be useful, especially when using less capable LLMs than DeepSeek-R1.
>
> > If there is a recent state-of-the-art (SOTA) method for the CASH problem, please mention it and compare your approach against it.
>
> The notion of "state-of-the-art" in the CASH problem is somewhat fluid because performance varies across benchmarks and evaluation settings. Nevertheless, one of the most recent and competitive methods is MaxUCB, which we already include in our comparisons.

---

### Official Review · Reviewer_h1kH · 2025-11-01

**Soundness:** 2
**Presentation:** 2
**Contribution:** 1
**Rating:** 4
**Confidence:** 2

**Summary:**

The paper frames CASH (algorithm + hyperparameters) as in-context meta-learning via prompting. Each dataset is summarized into a compact metadata block and an LLM is prompted to output model configs (chosen from a small family) in a single pass. Two modes: (i) Zero-Shot (only current task metadata) and (ii) Meta-Informed (prepend a few (metadata, best-config) exemplars from prior tasks) are proposed to get the final output. The method is compared against 4 other baseline methods and generally demonstrates improvements.

**Strengths:**

S1. Simple and practical setup: A Clean metadata template makes outputs trainable and reproducible.

S2. Single-pass: Gets a set of candidates at once without multiple prompting, easy to utilize.

S3. The performance on diverse types of tasks is promising.

**Weaknesses:**

W1. Mechanism ambiguity (meta-learning vs. knowledge bank: The evidence doesn’t separate genuine meta-learning from retrieval-ish pattern-matching. The paper just uses a structured task description as the context and lets the LLM figure out the details of the best hyperparameters. This just looks like an LLM being used as a knowledge bank, and a more comprehensive test requires an experiment to determine whether the LLM learns mappings rather than parrots "best" configs from memory.

W2. Narrow model/space coverage: The approach is restricted to four families and a fixed grid, which aids engineering but limits generality. The authors themselves note that extending coverage is essential and miss ablations on family set, grid granularity, K (ensemble size), and metadata richness make it hard to assess robustness.

**Questions:**

Refer Weakness.

---

> ### Author Response · Authors · 2025-11-20
> **Rebuttal by Authors**
>
> We thank Reviewer h1kH for their feedback and the time dedicated to evaluating our work.
>
> > W1. Mechanism ambiguity (meta-learning vs. knowledge bank: The evidence doesn’t separate genuine meta-learning from retrieval-ish pattern-matching. The paper just uses a structured task description as the context and lets the LLM figure out the details of the best hyperparameters. This just looks like an LLM being used as a knowledge bank, and a more comprehensive test requires an experiment to determine whether the LLM learns mappings rather than parrots "best" configs from memory.
>
> We agree that distinguishing meta-learning from pure memorization is an important concern. In our setting, however, the performance gains of the Meta-Informed approach over the Context-Random baselines provide evidence that the LLM is not simply retrieving memorized configurations. It suggests that the model is using the task-specific metadata to adapt its recommendations rather than parroting fixed "best" configurations. We are running an additional counterfactual experiment where the metadata–configurations pairs in the support set are deliberately shuffled. If performance degrades under this mismatch, this provides direct evidence that the LLM relies on the correct metadata–configuration alignment. We will include this analysis or discuss the results accordingly in the revised manuscript.
>
> > W2. Narrow model/space coverage: The approach is restricted to four families and a fixed grid, which aids engineering but limits generality. The authors themselves note that extending coverage is essential and miss ablations on family set, grid granularity, K (ensemble size), and metadata richness make it hard to assess robustness.
>
> Our goal in this paper is to show that LLMs can act as in-context meta-learners for the CASH problem. To isolate this capability, we focused on tabular tasks and a controlled set of four model families with a fixed configuration grid. This setup lets us evaluate the LLM’s behavior without the added complexity of a much larger or less structured search space. We agree that extending to broader model families, and richer metadata is important for assessing robustness.

---

### Author Response · Authors · 2025-11-20
**Rebuttal by Authors**

We thank all reviewers for their thoughtful feedback. We address here a general comment raised by all reviewers regarding the novelty of our approach and the limited number of model families considered. This work takes a first step toward leveraging pretrained LLMs as in-context meta-learners for more comprehensive CASH instances. Although the setup may resemble a standard application of in-context learning, the success was by no means guaranteed: it was not clear a priori that an LLM could infer effective model–hyperparameter configurations solely from dataset metadata. We also believe that demonstrating this capability has substantial potential impact: if LLMs can reliably perform such recommendations, this could significantly influence future applications of LLMs to hyperparameter recommendation, advance research on the CASH problem, and support developments within the AutoML community. To our knowledge, no prior work has investigated using LLMs as meta-learners to generate model–hyperparameter configurations directly from dataset metadata, which constitutes the central novelty of our approach. Our contribution is conceptual and empirical, providing a clear proof of concept: (i) an in-context meta-learning formulation of CASH, and (ii) empirical evidence that Meta-Informed prompting consistently outperforms Zero-Shot and non-LLM iterative baselines under the same training budget.

---

### Meta-Review · Area_Chair_vKD9 · 2026-01-06

**Summary:**

Reviewers raised consistent concerns regarding limited novelty, weak technical depth, and over-claimed framing as “in-context meta-learning.” Multiple reviewers noted that the approach is essentially a straightforward application of zero-/few-shot prompting to the CASH problem, without introducing new algorithms, learning mechanisms, or theoretical insights. The claimed “meta-learning” behavior is not convincingly distinguished from heuristic knowledge reuse or pattern matching.

Reviewers also highlighted restricted generality and evaluation limitations. The method is confined to four tabular model families and a fixed hyperparameter grid, with no ablations on model coverage, metadata design, ensemble size, or robustness. Empirical gains are mainly observed under very small training budgets, where traditional HPO methods are disadvantaged, and comparisons to iterative LLM-based HPO or stronger AutoML systems are missing. Several reviewers questioned the practicality of the Meta-Informed setting, which assumes access to high-quality prior task configurations obtained via prior search.

The rebuttal primarily provides clarifications and reiterates positioning but does not add new experiments or analyses that address these core concerns. As a result, the main issues around novelty, mechanism attribution, generality, evaluation fairness, and practical applicability remain largely unresolved, making it unlikely that reviewer scores would increase post-rebuttal.

**Reviewer Concerns:**

**Concerns largely addressed in the rebuttal**

- **Prompt order sensitivity** (Reviewer WDu3):
  The authors point out that Appendix H already evaluates robustness to prompt shuffling and reports no statistically significant effect, addressing this clarification-level concern.

- **Clarification of experimental details** (Reviewer WDu3):
  The rebuttal clarifies questions regarding the logistic regression baseline, search space definition, decoding behavior, and the non-monotonic scaling behavior across Qwen model sizes.

- **Scope clarification** (Reviewers h1kH–W2, LM2z–W1, 5AAH):
  The authors explicitly acknowledge the restricted focus on tabular tasks and four model families, reframing the contribution as a controlled proof-of-concept rather than a full CASH solution.

---

**Remaining limitations**

- **Lack of novelty / weak technical contribution** (Reviewers i4zb, h1kH–W1, WDu3, LM2z):
  The rebuttal reiterates conceptual positioning but does not resolve concerns that the work is a straightforward application of zero-/few-shot prompting, without introducing new algorithms, learning mechanisms, or theoretical insights.

- **Mechanism ambiguity: meta-learning vs. heuristic reuse or memorization** (Reviewer h1kH–W1, LM2z–W3):
  While the rebuttal argues that Meta-Informed gains indicate adaptation, no new experiments are provided to clearly distinguish genuine meta-learning from heuristic pattern matching or memorization.

- **Restricted generality and missing ablations** (Reviewers h1kH–W2, WDu3, LM2z–W1, 5AAH):
  Concerns regarding limited model families, fixed hyperparameter grids, metadata dependence, ensemble size, and robustness are acknowledged but remain empirically unaddressed, with no new ablation studies added.

- **Evaluation fairness and missing baselines** (Reviewer LM2z–W2, LM2z–W4):
  The rebuttal defends the low-budget comparison but does not add comparisons to iterative LLM-based HPO methods or stronger AutoML baselines, which remains a key concern.

- **Practicality of the Meta-Informed setting** (Reviewer i4zb, 5AAH):
  The assumption of access to high-quality prior task configurations obtained via prior search remains largely unresolved and continues to limit real-world applicability.

- **Potential pre-training contamination** (Reviewer LM2z–W5):
  The rebuttal argues that relative comparisons mitigate this issue, but no new evidence or experiments are provided to rule out dataset memorization effects.

**Reviewer Scores:**

- **Reviewer i4zb**: Original score 2 (reject).
  The rebuttal reiterates positioning but does not address the core concerns regarding lack of novelty, weak performance justification, and the impracticality of the Meta-Informed setting. No new experiments were added. The score would likely remain **2**.

- **Reviewer LM2z**: Original score 2 (reject).
  While the rebuttal responds point-by-point, major concerns about evaluation fairness, missing iterative LLM baselines, generality, and potential pre-training contamination remain unresolved, with no additional empirical evidence. The score would likely remain **2**.

- **Reviewer h1kH**: Original score 4 (borderline reject).
  The rebuttal clarifies scope and reiterates arguments regarding meta-learning vs. memorization, but does not add new experiments to resolve mechanism ambiguity or generality concerns. The score would likely remain **4**.

- **Reviewer WDu3**: Original score 4 (borderline reject).
  Clarification-level questions (e.g., prompt order sensitivity, experimental details) were addressed, but core concerns about novelty, lack of ablations, and missing baselines persist. No new analyses were added. The score would likely remain **4**.

- **Reviewer 5AAH**: Original score 4 (borderline reject).
  The rebuttal defends the Meta-Informed setting and limited model coverage conceptually, but does not address concerns about realism, generality across CASH, or missing ablations with new evidence. The score would likely remain **4**.

**Overall**, based on the rebuttal mainly providing clarifications and reframing without new experiments or analyses, a reasonable post-discussion interpretation of the reviews remains approximately **2, 2, 4, 4, and 4**, corresponding to a reject profile.

---

### Decision · Program_Chairs · 2026-01-26

Reject